



# A note on leveraging synergy in multiple meteorological datasets with deep learning for rainfall-runoff modeling

Frederik Kratzert[1], Daniel Klotz[1], Sepp Hochreiter[1], and Grey S. Nearing[2,3]

[1]LIT AI Lab & Institute for Machine Learning, Johannes Kepler University Linz, Austria
[2]Upstream Tech, Natel Energy Inc., Alameda, CA USA
[3]Department of Geological Sciences, University of Alabama, Tuscaloosa, AL United States

**Correspondence:** Frederik Kratzert (kratzert@ml.jku.at), Grey S. Nearing (grey@upstream.tech)

**Abstract.** A deep learning rainfall-runoff model can take multiple meteorological forcing products as inputs and learn to combine them in spatially and temporally dynamic ways. This is demonstrated using Long Short Term Memory networks (LSTMs) trained over basins in the continental US using the CAMELS data set. Using multiple precipitation products (NLDAS, Maurer, DayMet) in a single LSTM significantly improved simulation accuracy relative to using only individual precipitation

products. A sensitivity analysis showed that the LSTM learned to utilize different precipitation products in different ways in different basins and for simulating different parts of the hydrograph in individual basins.

## 1 Introduction

There are many different meteorological products that a hydrologist might choose as forcing data, and no data product is perfect. While temperature estimates between different forcing products are frequently similar, precipitation estimates are often subject

to large disagreements (e.g., Behnke et al., 2016; Timmermans et al., 2019). The appropriate choice of the input forcing data is an important step for every modelling task. To our knowledge it is so far not possible to dissect, which methodological choices lead to which disagreements in the data products (e.g., Beck et al., 2017; Newman et al., 2019); nor is it straightforward to estimate how these differences translate to model behavior (e.g., Yilmaz et al., 2005; Henn et al., 2018; Parkes et al., 2019). Thus, the choice of the *"right"* product, for a given modelling exercise, requires careful consideration.

Generally speaking, the most accurate precipitation data comes from in situ gauges, which are point-based measurements of complex spatial processes (although in certain cases, especially related to snow, modeled products might be better - e.g., Lundquist et al., 2019). Today's large-scale hydrological models, however, require data fields (usually gridded), which are model-based products, resulting from a combination of spatial interpolation and/or satellite retrieval algorithms. Each product is based on different sets of assumptions that each potentially introduce different types of error and information loss. As an

example, Behnke et al. (2016) showed that no existing gridded meteorological product is uniformly better than all others over the continental United States (CONUS).

In this context, we would like to point out that - depending of the goal of the modelling exercise - data-driven models can have an inherent advantage compared to traditional hydrological modeling techniques: *A single data-driven model can use multiple forcing products directly*. The models can learn to exploit potential synergies in different (imperfect) precipitation





data sets (or any other type of model input). In particular, deep learning models as used by Kratzert et al. (2019b, a) can take any number of different precipitation and other meteorological inputs at every timestep. Because the different input data sets are used simultaneously in a single nonparametric model, this has the potential to produce more accurate simulations by combining those inputs in spatiotemporally dynamic ways. The goal of this contribution is to test the strength of this hypothesis by assessing the model's ability to learn complex and spatiotemporally variable interactions between different precipitation

products.

## 2  Methods

### 2.1  Data

This study uses the Catchment Attributes and Meteorological dataset for Large Sample Studies (CAMELS; Newman et al., 2014; Addor et al., 2017b). CAMELS contains basin-averaged daily meteorological forcing inputs derived from three different

gridded data products for 671 basins across CONUS. The three forcing products are (i) DayMet (Thornton et al., 1997), (ii) Maurer (Maurer et al., 2002), and (iii) NLDAS (Xia et al., 2012), the former has 1 km x 1 km spatial resolution and the latter two have one-eighth degree spatial resolution. Although CAMELS includes 671 basins, to facilitate a direct comparison of results with previous studies we used only the subset of 531 basins that were originally chosen for model benchmarking by Newman et al. (2017), who removed all basins with area greater than 2000 km$^2$, and also all basins where there was a discrepancy of

more than 10% between different methods of calculating basin area. These 531 basins were used for all experiments in this study except benchmarking against traditional hydrology models (see Sect. 2.4.1), because the benchmark models are only available at 447 of the 531 basins.

Behnke et al. (2016) conducted a detailed analysis of eight different precipitation and surface temperature (daily max/min) data products, including the three used by CAMELS. Those authors compared gridded precipitation and temperature values to

station data using roughly 4000 weather stations across CONUS. Their findings were that *"no data set was 'best' everywhere and for all variables we analyzed"* and *"two products stood out in their overall tendency to be closest to (Maurer) and farthest from (NLDAS2) observed measurements."* Furthermore, they did not find a *"clear relationship between the resolution of gridded products and their agreement with observations, either for average conditions ... or extremes"* and noted that the *"high-resolution DayMet ... data sets had the largest nationwide mean biases in precipitation."*

Figure 1 gives an example of disagreement between precipitation products in CAMELS that we hope to capitalize on by training a model with multiple forcing inputs. This figure shows the noisy relationship between the three precipitation products in a randomly-selected basin (USGS ID: 07359610). Deep Learning approaches can learn to mitigate the type of noise shown in the scatter plot in the right-hand panel of Fig. 1 and use the inherent information by using multiple forcing products simultaneously in a single model.

The left-hand subplot of Fig. 1 shows a time-shift between DayMet and Maurer precipitation in the same basin. This type of shift is common. Behnke et al. (2016), for example, reported that *"[b]ecause gridded products differ in how they define a calendar day (e.g., local time relative to Coordinated Universal Time), appropriate lag correlations were applied through*





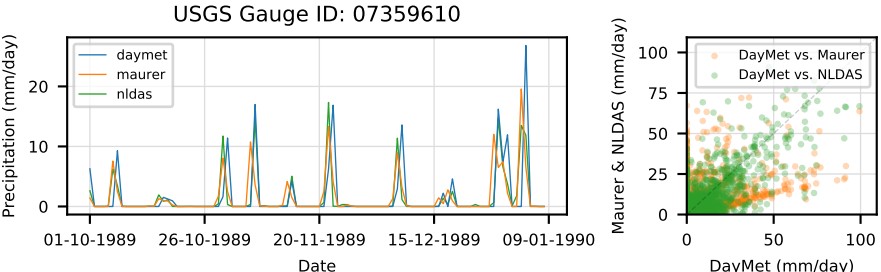

**Figure 1.** Illustration of the relationship between three CAMELS precipitation products at a randomly-selected basin. The left-hand subplots show the first 100 days of precipitation data from all three products during the test period, and the right-hand subplot shows scatter between the three products over the full test period. The scatter shown in the right-hand subplot is the data uncertainty that we would like to mitigate by using multiple forcings simultaneously in a deep learning rainfall-runoff model. In this particular basin, there appears to be a 1-day shift between DayMet and Maurer, which is common in the CAMELS data set (this shift is apparent in 325 of the 531 basins; see Figure 2)

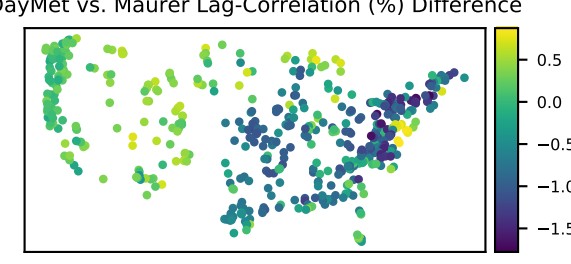

**Figure 2.** Spatial distributions of lagged vs. non-lagged correlations between DayMet and Maurer test-period precipitation. Positive values indicate that the 1-day lagged correlation is higher.

*cross- correlation analysis to account for the several- hour offset in daily station data."* We performed a lag-correlation analysis on the precipitation products in CAMELS and found a higher correlation between DayMet and Mauer when Mauer was lagged
by one day in 325 (of 531) basins. Figure 2 shows the percent difference between lagged vs. non-lagged correlations between DayMet and Maurer.

Each of the forcing products in CAMELS includes daily precipitation (mm/d) and maximum and minimum daily temperature (°C), vapor pressure (Pa), and surface radiation (W/m$^2$). The original CAMELS data set hosted by the US National Center for Atmospheric Research (Newman et al., 2014) only contains daily mean temperatures for Maurer and NLDAS. CAMELS-
relevant Maurer and NLDAS products with daily minimum and maximum temperatures are available from our HydroShare DOI (see data availability section). In addition to the three daily forcing data sets from CAMELS, we used the same 27 catchment attributes as Kratzert et al. (2019a, b), which consist of topological, climatic, vegetation, and soil descriptors (Addor





et al., 2017a). Prior to training any model, all input variables were normalized independently by subtracting the CONUS-wide mean and dividing by the CONUS-wide standard deviation.

## 2.2 Models

Long Short-Term Memory networks (LSTMs) are a type of recurrent neural network (Hochreiter, 1991; Hochreiter and Schmidhuber, 1997b; Gers et al., 1999). LSTMs are a type of state-space model that function through a set of input-state-output relationships. Gates, which are activated linear functions, control information flows from inputs and previous states to current state values (called an input gate), from current states to outputs (called an output gate), and also control the timescale of each element of the state vector (called a forget gate). States (usually called cell states) accumulate and store information over time, much like the states of a dynamical system. Technical details of the LSTM model architecture have been described in several previous publications in hydrology journals, and we refer the reader to Kratzert et al. (2018) for a detailed explanation geared towards hydrologists.

## 2.3 Experimental Design

To conduct our analyses we trained an LSTM model using each of the three forcing products together, separate LSTM models for each pairwise combination of forcing products (DayMet & Maurer, DayMet & NLDAS, and Maurer & NLDAS), and separate LSTMs for all three forcing products individually.

For each of these seven input configurations, we trained an ensemble of $n = 10$ different LSTMs with different randomly initialized weights. We report the statistics from averaging the simulated hydrographs from each of these 10-member ensembles (single model results are provided in the Appendix). Ensembles are used to account for randomness inherent in the training procedure. The importance of using ensembles for this purpose was demonstrated by Kratzert et al. (2019b).

We used the same time periods for model training and testing as by Kratzert et al. (2019b) - this allows us to directly compare results of this study with the full set of benchmark hydrology models used by that previous study. The training period was from 1 October 1999 to 30 September 2008 (9 years of training data for each catchment) and the test period was 1 October 1989 to 30 September 1999 (10 years of test data for each catchment).

Similar to previous studies (Kratzert et al., 2019b, a), we used LSTMs with 256 memory cells and a dropout rate of 0.4 (40%) in the fully connected layer that derives network predictions (streamflow) from LSTM output. All models were trained with a mini-batch size of 256 for 30 epochs using the Adam optimizer (Kingma and Ba, 2014) with an initial learning rate of 1e-3, reduced to 5e-4 after 20 epochs and further reduced to 1e-4 after 25 epochs. All inputs were standardized to have zero mean and unit variance over all 531 catchments collectively. During model evaluation, negative predictions in the original value space were clipped to zero, i.e. no negative discharges. The loss function was the basin-averaged Nash-Sutcliffe Efficiency (NSE), see Kratzert et al. (2019b).



## 2.4 Analysis

We examined the experiments described above with three types of analysis. The goal is to provide different illustrations of how
the LSTM leveraged multiple forcing products.

- **Analysis 1 - Feature Ablation**: An *ablation study* removes parts of the network to gain a better understanding of the model. We adopted this procedure by removing the different meteorological forcing products in a step-wise fashion and comparing the individual results by using multiple performance metrics and signatures (see Table 1). To provide context we also compare them against a family of conceptual and process-based hydrological models.

- **Analysis 2 - Precipitation Uncertainty**: We used triple collocation to estimate spatially-varying error characteristics of the three precipitation forcing products, and assessed relationships between these error statistics with the performance of both single- and multiple-forcing LSTMs. These experiments help us understand where we can expect value from using multiple forcing products in a single model.

- **Analysis 3 - Sensitivity & Contribution**: We performed an input attribution analysis of the trained LSTM models to quantify how the LSTM learned to leverage different forcing products in spatiotemporally dynamic ways.

### 2.4.1 Analysis 1: Feature Ablation

All LSTM ensembles were trained using a squared-error loss function (the basin-averaged NSE), however we are interested to know how the models simulate different aspects of the hydrograph. As such, we report a collection of hydrologically-relevant performance metrics outlined in Table 1. These statistics include the standard time-average performance metrics (NSE, KGE), as well as comparisons between observed and simulated hydrologic signatures. The hydrologic signatures we report are the same ones used by Addor et al. (2018). For each signature, we compute the Pearson correlation between the signature derived from the observed discharge and derived from the simulated discharge of all basins.

To provide a baseline for comparison, LSTM ensembles were benchmarked against the same family of hydrological models used for benchmarking by Kratzert et al. (2019b). These models are: (i) SAC-SMA (Burnash et al., 1973; Burnash, 1995) coupled with the Snow-17 snow routine (Anderson, 1973), hereafter referred to as SAC-SMA, (ii) VIC (Liang et al., 1994), (iii) FUSE (Clark et al., 2008; Henn et al., 2008) (three different model structures, 900, 902, 904), (iv) HBV (Seibert and Vis, 2012), and (v) mHM (Samaniego et al., 2010; Kumar et al., 2013). Some of these models were calibrated to individual basins and others were regionally calibrated. All of the benchmarks used Maurer forcings and all were calibrated and validated on the same time periods used in this study. In order to avoid any potential or implicit bias, we did not run any of our own benchmark models - all models were solicited originally by Kratzert et al. (2019b) from different groups with experience running each individual model. The whole family of benchmark model runs is only available in 447 of the 531 CAMELS catchments, chosen by Newman et al. (2017). Thus, while we use the set of 531 catchments for all other parts of this study, we only considered 447 catchments for benchmarking against traditional hydrology models.





**Table 1.** Description of the performance metrics (top part) and signatures (bottom part) considered in this study. For each signature, we derived a metric by computing the Pearson correlation between the signature of the observed flow and the signature of the simulated flow over all basins. Description of the signatures taken from Addor et al. (2018)

| Metric/Signature | Description | Reference |
|---|---|---|
| NSE | Nash-Sutcliff efficiency | Eq. 3 in Nash and Sutcliffe (1970) |
| KGE | Kling-Gupta efficiency | Eq. 9 in Gupta et al. (2009) |
| Pearson rt | Pearson correlation between observed and simulated flow | |
| $\alpha$-NSE | Ratio of standard deviations of observed and simulated flow | From Eq. 4 in Gupta et al. (2009) |
| $\beta$-NSE | Ratio of the means of observed and simulated flow | From Eq. 10 in Gupta et al. (2009) |
| FHV | Top 2% peak flow bias | Eq. A3 in Yilmaz et al. (2008) |
| FLV | Bottom 30% low flow bias | Eq. A4 in Yilmaz et al. (2008) |
| FMS | Bias of the slope of the flow duration curve between the 20% and 80% percentile | Eq. A2 Yilmaz et al. (2008) |
| Peak-Timing | Mean peak time lag (in days) between observed and simulated peaks | See Appendix C |
| Baseflow index | Ratio of mean daily baseflow to mean daily discharge | Ladson et al. (2013) |
| HFD mean | Mean half-flow date (date on which the cumulative discharge since October first reaches half of the annual discharge) | Court (1962) |
| High flow dur. | Average duration of high-flow events (number of consecutive days >9 times the median daily flow) | Clausen and Biggs (2000), Table 2 in Westerberg and McMillan (2015) |
| High flow freq. | Frequency of high-flow days (>9 times the median daily flow) | Clausen and Biggs (2000), Table 2 in Westerberg and McMillan (2015) |
| Low flow dur. | Average duration of low-flow events (number of consecutive days <0.2 times the mean daily flow) | Olden and Poff (2003), Table 2 in Westerberg and McMillan (2015) |
| Low flow freq. | Frequency of low-flow days (<0.2 times the mean daily flow) | Olden and Poff (2003), Table 2 in Westerberg and McMillan (2015) |
| Q5 | 5% Flow quantile (low flow) | |
| Q95 | 95% Flow quantile (high flow) | |
| Q mean | Mean daily discharge | |
| Runoff ratio | Runoff ratio (ratio of mean daily discharge to mean daily precipitation, using DayMet precipitation) | Eq. 2 in Sawicz et al. (2011) |
| Slope FDC | Slope of the flow duration curve (between the log-transformed 33rd and 66th streamflow percentiles | Eq. 3 in Sawicz et al. (2011) |
| Stream elasticity | Streamflow precipitation elasticity (sensitivity of streamflow to changes in precipitation at the annual time scale, using DayMet precipitation) | Eq. 7 in Sankarasubramanian et al. (2001) |
| Zero flow freq. | Frequency of days with zero discharge. | |

### 2.4.2 Analysis 2: Precipitation Uncertainty

The objective of the second analysis is to demonstrate that the multiple-forcing model learns to leverage patterns in forcing data error structures. Our approach was to relate error characteristics of the different precipitation products with model performance, and with performance improvements due to using multiple forcing products. We used triple collocation to estimate error characteristics of the different forcing products. Triple collocation is a statistical technique to estimate error variances of three or more noisy measurement sources without knowing the true values of the measured quantities (Stoffelen, 1998; Scipal et al.,

2010). Its major assumptions are that the error models are linear and independent between sources; in particular, that all (three or more) measurement sources are each a combination of a scaled value of the true variable plus some additive random noise:

$$M_{i,t} = \alpha_i T_t + \varepsilon_{i,t}, \tag{1}$$





where $M_*$ are measurement values (i.e. here the modeled precipitation values), subscript $i$ represents the source (DayMet, Maurer, NLDAS), and subscript $t$ represents the timestep in the test period (1 October 1989 to 30 September 1999); $T_*$ is the

unobserved true value of total precipitation in a given catchment on a given day; $\varepsilon_*$ are i.i.d. measurement errors from any distribution.

The linearity assumption is not appropriate for precipitation data, which are typically assumed to have multiplicative error. Following Alemohammad et al. (2015), we assumed a multiplicative error model for all three precipitation source, and converted these to linear error models by working with the log-transformed precipitation data:

$$M_{i,t} = \alpha_i T_t{}^{\beta_i} + e^{\varepsilon_{i,t}} \qquad (2)$$

$$\ln(M_{i,t}) = \alpha_i + \beta_i T_t + \varepsilon_{i,t}. \qquad (3)$$

Standard triple collocation is then applied, so that estimates of the error variances for each source are:

$$\sigma_i = C_{i,i} - \frac{C_{i,j} C_{i,k}}{C_{j,k}}, \qquad (4)$$

for all $i, j, k$, where $C_{i,j}$ is the covariance between the time series of source $i$ and source $j$; $\sigma_i$ is the variance of the distribution

that each i.i.d. $\varepsilon_{i,t}$ is drawn from.

Additionally, extended triple collocation (McColl et al., 2014) allows us to derive the correlation coefficients between measurement sources and truth as:

$$\rho_i = \frac{C_{i,j} C_{i,k}}{C_{i,i} C_{j,k}}. \qquad (5)$$

This triple collocation analysis was applied separately in each of the 531 CAMELS catchments to obtain basin-specific

estimates of the error variances, $\sigma_i$, and truth-correlations, $\rho_i$, for each of the three precipitation products. Albeit the assumption that the forcing products have independent error structures (i.e. $\varepsilon_{i,t} \perp\!\!\!\perp \varepsilon_{j,t}$) is not met in our case we expect the results to be robust enough for the purpose at hand.

### 2.4.3   Analysis 3: Sensitivity & Contribution

All neural networks (like LSTMs) are differentiable *almost everywhere* by design. Therefore, a gradient-based contribution

analysis seems natural. However, as discussed by Sundararajan et al. (2017), the naive solution of using local gradients is not a reliable measures of sensitivity, since gradients might be flat even if the model response is heavily influenced by a particular input data source (which is not by necessity a bad property, see for example Hochreiter and Schmidhuber, 1997a). This is especially true in neural networks, where activation functions often include step-changes over portions of the input space - e.g., the sigmoid and hyperbolic tangent activation functions used by LSTMs have close-to-zero gradients at both extremes (see

also: Shrikumar et al., 2016; Sundararajan et al., 2017).

Sundararajan et al. (2017) proposed a method of input attribution for neural networks which accounts for this described lack of local sensitivity: *integrated gradients*. Integrated gradients are a path integral of the gradients from some baseline input $x'$,





to the actual value of the input, $x$:

$$\text{IntegratedGrads}_i^{\text{approx}}(\boldsymbol{x}) := \frac{\mathbf{x}_i - \mathbf{x'}_i}{m} \sum_{k=1}^{m} \frac{\partial F(\tilde{\mathbf{x}})}{\partial \tilde{\mathbf{x}}_i} \Bigg|_{\tilde{\mathbf{x}} = \mathbf{x'} + \frac{k}{m}(\mathbf{x} - \mathbf{x'})}. \tag{6}$$

We used a value of zero precipitation everywhere as the baseline for calculating integrated gradients with respect to the three different precipitation forcings (DayMet, Maurer, NLDAS). We calculated the integrated gradients of each daily streamflow estimate in each CAMELS basin during the 10-year test period with respect to precipitation inputs from the past 365 days (the look-back period of the LSTM). That is, on day $t = T$, we calculated $1095 = 3 * 365$ integrated gradient values related to the three precipitation products. The relative integrated gradient values quantify how the LSTM combines precipitation products

over time, over space, and also as a function of lag or lead-time into the current streamflow prediction. In theory, one has to take into account the effect of *"explaining away"*, when analysing the decision process in models (Pearl, 1988; Wellman and Henrion, 1993). However, we assume that if evaluated over hundreds of basins and thousands of time steps, this effect is largely averaged out and therefore the analysis provides an indication of the actual information used by the model.

## 3   Results & Discussion

### 3.1   Results: Analysis 1 - Feature Ablation

The feature ablation analysis compared NSE values over 10-year test periods from the CAMELS basins for the seven distinct input combinations. As shown in Fig. 3, the three-forcing model had a median NSE value of 0.82 for the 447 basins, which were available for all benchmarking models. The three-forcing model outperformed all two-forcing models. Similarly, all two-forcing models outperformed all single-forcing models (all improvements were statistically significant at $\alpha = 0.001$, using

the Wilcoxon test). The best single-forcing LSTM had a median NSE of 0.76. This indicates that the LSTM was able to leverage unique information in the precipitation signals (this is not an unusual finding in the context of machine learning, see for example: Sutton, 2019). We also note that the single-forcing LSTM with Maurer inputs outperformed the single-forcing NLDAS model, which agrees with the results of Behnke et al. (2016) who showed that Maurer precipitation was generally more accurate than NLDAS precipitation.

To put these results into context, Fig. 4 compares all LSTMs against the benchmark hydrology models in the 447 basins where simulations of all benchmark models were available. All LSTM models were better than all benchmark models through the entire CDF curve. As a point of reference, the difference in the median NSE between the best-performing single-forcing LSTM (DayMet) and the best-performing traditional hydrology model (HBV) was 0.09, while using all three CAMELS forcings increased that improvement over traditional models by another 0.055 (61%). The total improvement in the median NSE

of the multi-forcing LSTM over the best-performing hydrology model was 0.143 (21%).

Table 2 and Table 3 give the benchmarking results from all metrics and signatures in Table 1. The three-forcing LSTM out-performs all benchmark models against all metrics except $\beta$-NSE decomposition, the bias of the slope of the flow duration curve (FMS) and the bias of the low flows (FLV). The three-forcing LSTM also out-performs all benchmark models against all





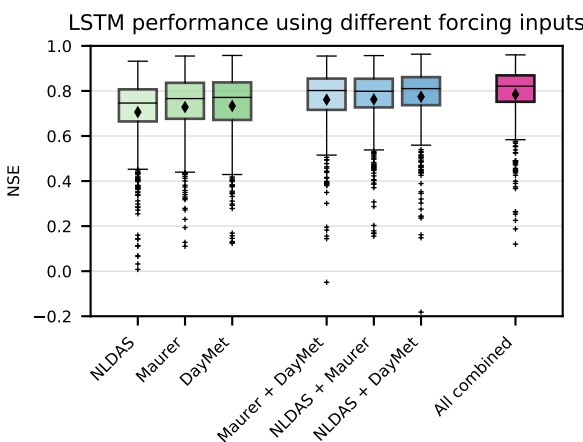

**Figure 3.** Test-period comparison between single-forcing and multiple-forcing LSTM ensembles ($n = 10$) over 447 CAMELS basins. All differences are statistically significant ($\alpha = 0.001$), with the exceptions of "DayMet" vs. "Maurer" ($p \approx 0.08$) and "NLDAS + Maurer" vs. "Maurer + DayMet" ($p \approx 0.4$)

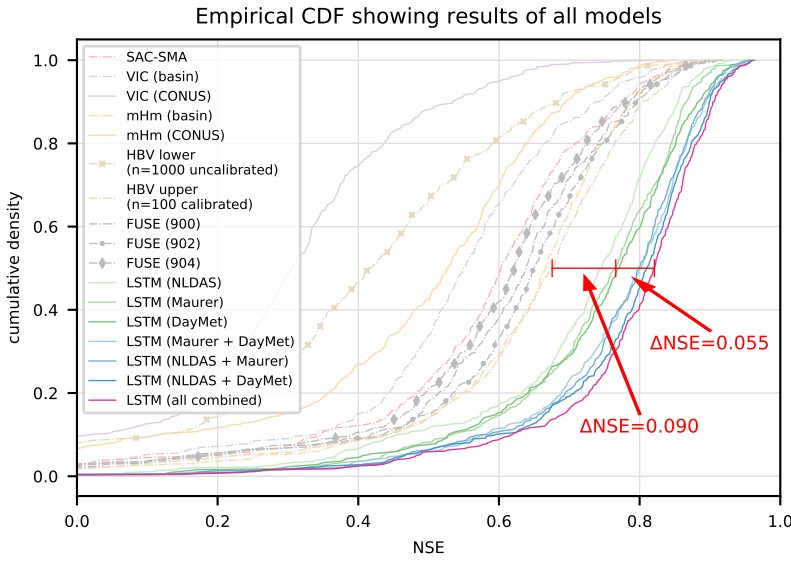

**Figure 4.** Empirical cumulative density function of the NSE performance over the 447 commonly modelled basins of all LSTMs and benchmark models. The red indicator lines mark the median NSE difference between the best hydrological benchmark model (ensemble of 100 calibrated HBV models) and the LSTM of our previous publication (Kratzert et al., 2019b), as well as the current best, if trained with all forcing products combined.

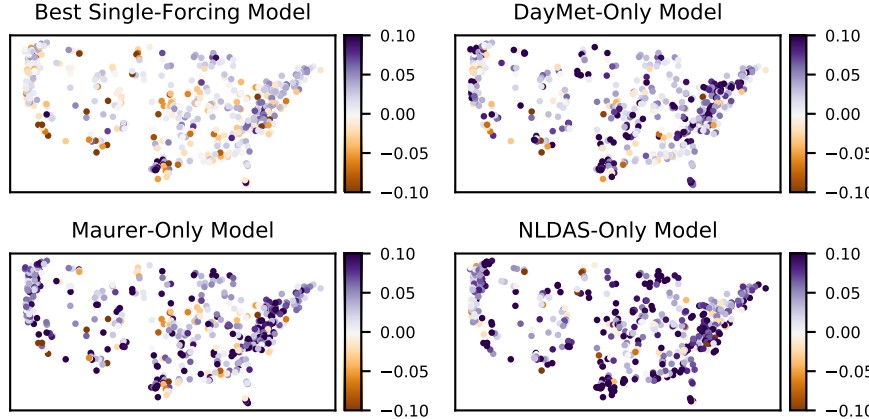

**Figure 5.** Spatial distribution of the NSE differences between the three-forcing LSTM relative to the best single-forcing model in each basin (top-left subplot), and relative to each single-forcing model (other three subplots). The three-forcing LSTM was better than the best single-forcing model in 351 of 531 basins (66%) and was better than each single-forcing model in: 443 (83%; DayMet), 456 (86%; Maurer), and 472 (89%; NLDAS) basins.

hydrologic signatures except the ones related to low-flows (frequency of zero flows and frequency and duration of flows below
20% of basin-average). We therefore note that the LSTM approach - while generally providing the best available model - still
has approximation difficulties towards the extreme lower-end of the runoff distribution.

Looking at the spatial distribution of the performance differences in all basins used for model training (i.e., 531 basins instead
of the 447 basins used for the benchmarking described above), it is evident that the three-forcing LSTM outperformed the
single forcing models almost everywhere (Fig. 5). Individual exceptions where *"less is more"* do however exist (e.g. Southern
California). Concretely, the three-forcing model was better than the best single forcing model in 66% of the basins (351 of 531)
and had a higher NSE than the individual single-forcing LSTMs in over 80% of the basins

## 3.2   Results: Analysis 2 - Precipitation Uncertainty

DayMet typically produces lower NSE values in basins where triple collocation reported that the DayMet precipitation error
variances are high. This is what we would expect: low skill in basins with high precipitation error; however we did not see
similar patterns with the other two precipitation products (see Fig. 6, where the triple collocation error variances and truth-
correlation are plotted against the NSE scores of the single-source models) . In fact, the NLDAS LSTM tends to perform worse
in basins with lower precipitation error (as estimated by triple collocation).

Figure 7 is an adapted version of Fig. 6 that highlights a few high-skill, high-variance NLDAS basins in blue. These basins
correspond to a cluster of basins in the Rocky Mountains (Fig. 8) where NLDAS has low correlation with the other two products
but still yields high-skill LSTM simulations.





**Table 2.** Values of the benchmarking metrics from Table 1. Bold indicates the best value per metric or signature. Multiple bold numbers per row mean that there is no statistical difference to the best performing model in the given metric/signature at ($\alpha = 0.001$).

| Metric | LSTM (all) | SAC-SMA | VIC (basin) | VIC (CONUS) | mHM (basin) | mHM (CONUS) | HBV (lower) | HBV (upper) | Fuse (900) | Fuse (902) | Fuse (904) |
|---|---|---|---|---|---|---|---|---|---|---|---|
| NSE[i] (median) | **0.82** | 0.60 | 0.55 | 0.31 | 0.67 | 0.53 | 0.42 | 0.68 | 0.64 | 0.65 | 0.62 |
| NSE[i] (mean) | **0.79** | 0.56 | 0.52 | 0.17 | 0.63 | 0.44 | 0.24 | 0.63 | 0.59 | 0.61 | 0.58 |
| KGE[ii] | **0.81** | 0.63 | 0.59 | 0.26 | 0.69 | 0.47 | 0.39 | 0.68 | 0.67 | 0.69 | 0.64 |
| Pearson r[iii] | **0.92** | 0.79 | 0.76 | 0.65 | 0.83 | 0.79 | 0.71 | 0.83 | 0.82 | 0.82 | 0.81 |
| α-NSE[iv] | **0.87** | 0.78 | 0.73 | 0.46 | 0.81 | 0.59 | 0.58 | 0.79 | 0.80 | 0.80 | 0.78 |
| β-NSE[v] | -0.03 | -0.07 | **-0.02** | -0.07 | -0.04 | -0.04 | -0.02 | **-0.01** | -0.03 | -0.05 | -0.07 |
| FHV[vi] | **-13.32** | -20.36 | -28.14 | -56.48 | -18.64 | -40.18 | -41.86 | -18.49 | -18.94 | -19.36 | -21.41 |
| FLV[vii] | 40.81 | 37.42 | -74.77 | 18.87 | 11.43 | 36.80 | 23.88 | 18.34 | **-10.54** | -68.22 | -67.60 |
| FMS[viii] | -8.15 | -14.36 | **-6.56** | -27.99 | -7.22 | -30.35 | -15.94 | -24.94 | **-5.09** | 9.75 | 15.49 |
| Peak-Timing[ix] | **0.36** | 0.81 | 0.69 | 0.92 | 0.69 | 0.75 | 1.21 | 0.63 | 0.57 | 0.57 | 0.61 |

*[i]: Nash-Sutcliffe efficiency: $(-\infty, 1]$, values closer to one are desirable.*
*[ii]: Kling-Gupta efficiency: $(-\infty, 1]$, values closer to one are desirable.*
*[iii]: Pearson correlation: $[-1, 1]$, values closer to one are desirable.*
*[iv]: α-NSE decomposition: $(0, \infty)$, values close to one are desirable.*
*[v]: β-NSE decomposition: $(-\infty, \infty)$, values close to zero are desirable.*
*[vi]: Top 2 % peak flow bias: $(-\infty, \infty)$, values close to zero are desirable.*
*[vii]: 30 % low flow bias: $(-\infty, \infty)$, values close to zero are desirable.*
*[viii]: Bias of FDC midsegment slope: $(-\infty, \infty)$, values close to zero are desirable.*
*[ix]: Lag of peak timing: $(-\infty, \infty)$, values close to zero are desirable.*



**Table 3.** Values of the correlation coefficients (over 447 basins) of the simulated *vs.* observed hydrological signatures from Table 1. Bold indicates that the model is not statistically different than the best performing model in a given metric.

| Signature | LSTM (all) | SAC-SMA | VIC (basin) | VIC (CONUS) | mHM (basin) | mHM (CONUS) | HBV (lower bound) | HBV (upper bound) | Fuse (900) | Fuse (902) | Fuse (904) |
|---|---|---|---|---|---|---|---|---|---|---|---|
| Baseflow index | **0.92** | 0.84 | 0.75 | 0.29 | 0.78 | 0.30 | 0.17 | 0.56 | 0.70 | 0.87 | 0.80 |
| HFD mean | **0.98** | 0.92 | 0.92 | 0.87 | 0.95 | 0.91 | 0.89 | 0.94 | 0.93 | 0.95 | 0.94 |
| High flow dur. | **0.88** | 0.72 | 0.60 | 0.51 | 0.71 | 0.73 | 0.74 | 0.76 | 0.62 | 0.76 | 0.67 |
| High flow freq. | **0.85** | 0.67 | 0.66 | 0.43 | 0.63 | 0.52 | 0.40 | 0.55 | 0.60 | 0.75 | 0.71 |
| Low flow dur. | 0.48 | 0.30 | 0.31 | 0.23 | 0.39 | 0.38 | 0.40 | **0.55** | 0.32 | 0.48 | 0.34 |
| Low flow freq. | 0.79 | 0.75 | 0.63 | 0.26 | 0.61 | 0.33 | 0.20 | 0.32 | 0.58 | **0.84** | 0.71 |
| Q5 | **0.96** | 0.81 | 0.74 | 0.42 | 0.81 | 0.64 | 0.73 | 0.77 | 0.68 | 0.86 | 0.61 |
| Q95 | **0.99** | 0.98 | 0.97 | 0.90 | 0.98 | 0.93 | 0.93 | 0.98 | 0.98 | 0.98 | 0.98 |
| Q mean | **1.00** | 0.98 | 0.98 | 0.95 | 0.98 | 0.95 | 0.96 | 0.99 | 0.98 | 0.99 | 0.98 |
| Runoff ratio | **0.99** | 0.96 | 0.94 | 0.83 | 0.94 | 0.83 | 0.87 | 0.96 | 0.94 | 0.96 | 0.94 |
| Slope FDC | **0.67** | 0.61 | 0.62 | 0.44 | 0.49 | 0.48 | 0.49 | 0.42 | 0.45 | 0.65 | 0.59 |
| Stream elasticity | **0.75** | 0.66 | 0.51 | 0.31 | 0.66 | 0.57 | 0.57 | 0.57 | 0.55 | 0.66 | 0.62 |
| Zero flow freq. | 0.02 | **0.46** | 0.36 | 0.10 | -0.00 | NaN | NaN | NaN | NaN | 0.01 | NaN |

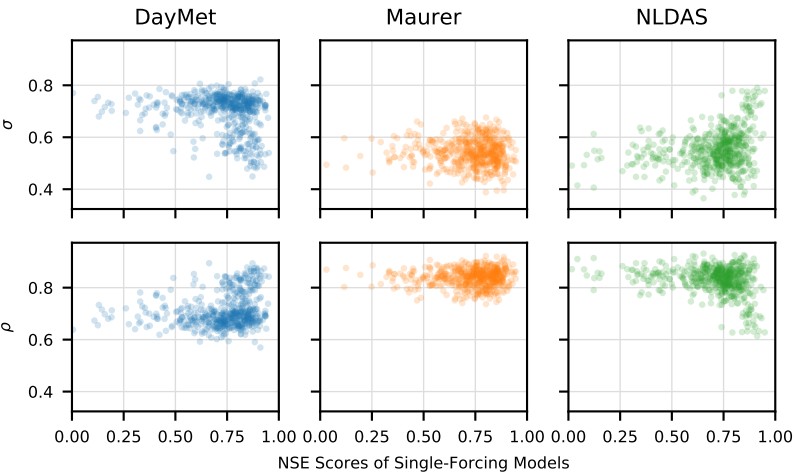

**Figure 6.** Triple collocation error variances and truth-correlations plotted against NSE scores of the single-forcing LSTM models. DayMet typically produces lower NSE values in basins where triple collocation reports that the precipitation error variances are high, whereas NLDAS produces lower NSE values in basins where triple collocation reports that the error variances are low. There is no apparent pattern in the Maurer data.

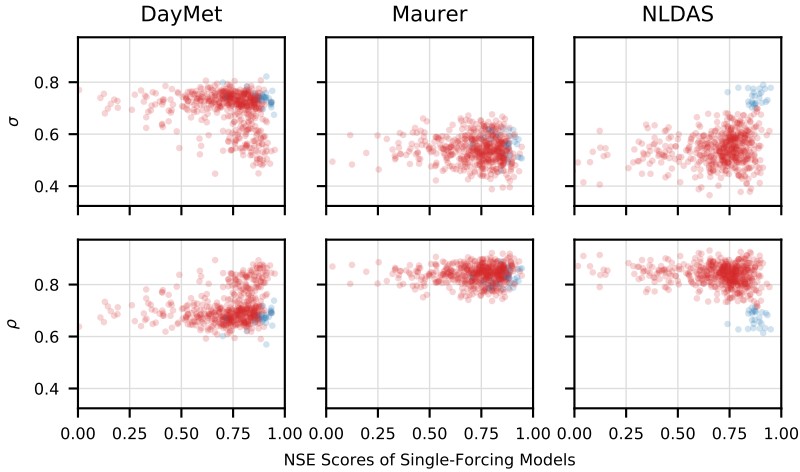

**Figure 7.** As in Fig. 6 the triple collocation error variances and truth-correlations are plotted against NSE scores of the single-forcing LSTM models. The coloring shows the anomalous NLDAS basins in blue and all others in red. For these basins NLDAS has low correlation with the other two products but still yields high-skill simulations.

Triple collocation measures (dis)agreement between measurement sources, rather than error variances directly. Figure 9 plots model performance against the individual variances of the precipitation products in each basin. This figure shows that the single-forcing DayMet LSTM tends to perform better in catchments with higher total precipitation variance (not triple





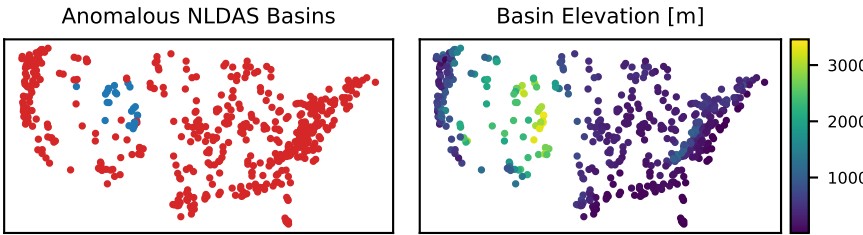

**Figure 8.** Spatial distribution of anomalous NLDAS basins shown in Fig. 7 (left) compared with elevation of the CAMELS basins (right).

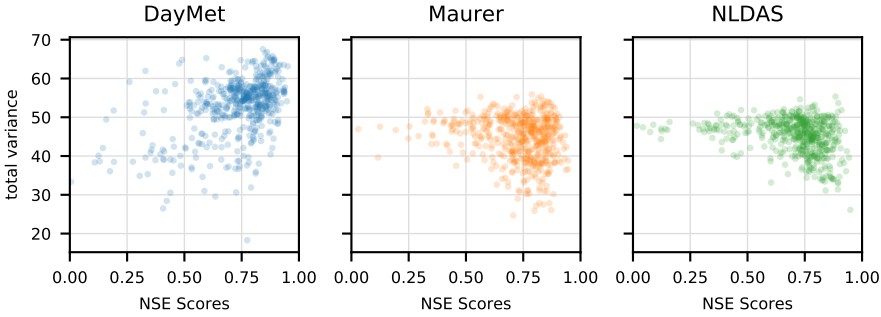

**Figure 9.** Performance of single-input models relative to the total variance of log-precipitation in each basin. The DayMet model tends to perform better in wetter basins (as the total DayMet variance increases), but the other two products have poor performing basins in catchments with high precipitation variance.

collocation error variance), indicating better performance in wetter catchments. This is not true for the other two models, where higher total variance is associated with a higher variance in model skill, indicating higher proportion of the variance due to measurement error.

To analyse the synergy due to using all forcings in a single LSTM we transposed the NSE *improvements* in each basin (due to using all three forcing products in the same LSTM) with the log-determinant of the covariance matrix of all three (standardized, log-transformed) precipitation products (Fig. 10). The log-determinant is a proxy for the joint entropy of the three (standardized, log-transformed) products, and increases when there is larger disagreement between the three data sets. Unlike in Fig. 9, the variances in Fig. 10 were calculated after removing the mean and overall variance of each log-transformed precipitation product so that the log-determinant of the covariance is not affected by the overall magnitude of precipitation in each catchment (i.e., does not increase in wetter catchments). With the exception of the anomalous NLDAS basins, Fig. 10 shows that the three-forcing model offers improvements with respect to the single-forcing models when there is larger disagreement between the three data sets.



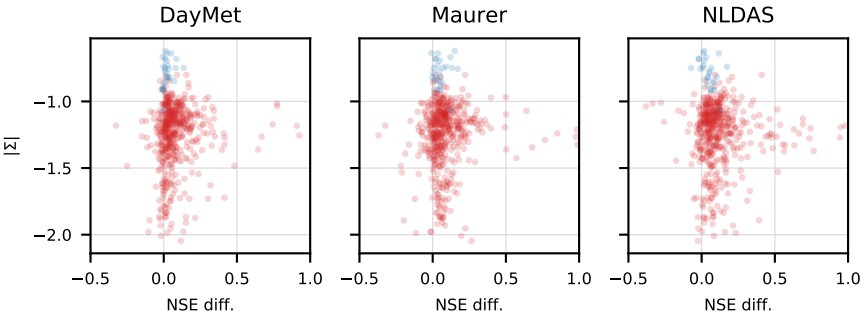

**Figure 10.** Fractional increase in NSE from the three-forcing model relative to the single-forcing models plotted against the log-determinant of the covariance matrix of all three (standardized, log-transformed) precipitation products. With the exception of the anomalous NLDAS basins (blue markers), the three-forcing model offers improvements with respect to the single-forcing models when there is larger disagreement between the three data sets. The three-forcing model learned to leverage synergy in these three precipitation products.

### 3.3 Results: Analysis 3 - Sensitivity & Contribution

Figure 11 shows the time- and basin-averaged integrated gradient of the multi-forcing LSTM as a function of lead time. To reiterate from above, the integrated gradient is a measure of input attribution, or sensitivity such that inputs with higher integrated gradients have a larger influence on model outputs. Integrated gradients shown in Fig. 11 were averaged over all timesteps in the test period, and also over all basins. This figure shows the sensitivity of streamflow at time $t = T$ to each of the three precipitation inputs at times $t = T - s$ where $s$ is the lag value on the x-axis. The main takeaways from this high-level illustration of the input sensitivities are: (1) that the sensitivity of current streamflow to precipitation decays with lead time (i.e., time before present) and (2) that the multi-forcing model has learned to ignore the Maurer input at the present timestep. The reason for the latter is the time shift in the Maurer product illustrated in Fig. 2.

The multi-forcing LSTM learned to combine the different precipitation products in spatiotemporally variable ways. Fig. 11 demonstrates the overall behavior of the multi-forcing LSTM. It is, however a highly condensed aggregate of a highly non-linear system. As such a lot of specific information is lost - as is always the case when nonlinearities are aggregated.

Therefore, Fig. 12 details the overall model behavior (through the lense of integrated gradients) by basin, and up to a lead time of $s = 3$ days prior to present. The model largely ignores Maurer precipitation at the current timestep in most basins - as is already apparent in Fig. 11, but the ratio of the contributions of each product varies between basin. Figure 12 shows relative contributions of each precipitation product, but the overall importance of precipitation also varies between basin.

Similarly, Fig. 13 shows the spatial extend of the most sensitive contribution over all time steps (left-subplot) and the the overall sensitivity to all three products combined (right-subplot), which is highly correlated with the total (or average) precipitation in the basin.. That is, they display the sum of the integrated gradients over time, lag, and product. From the right-subplot it becomes evident that the precipitation has a larger contribution to the sensitivity of streamflow predictions in wetter basins. Figure 13 also shows the product with the highest overall contribution in each basin.



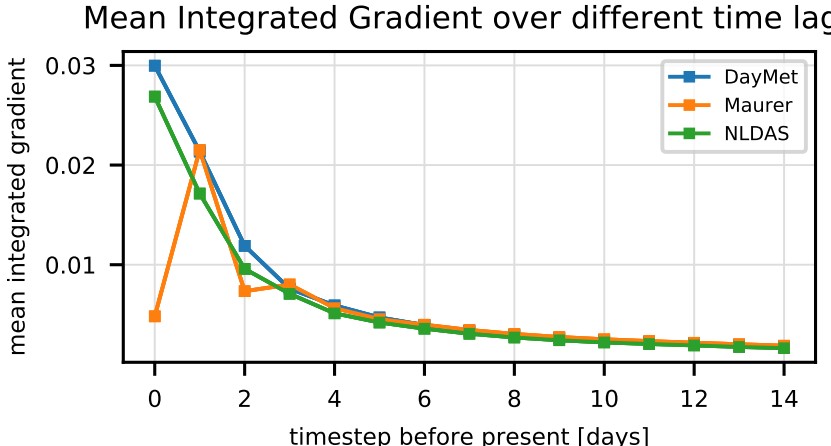

**Figure 11.** Time- and basin-averaged integrated gradients as a function of lag time (days before current streamflow prediction) of the three precipitation products. Because of the time shift shown in Fig. 2, the model has learned to ignore the Maurer input at the current timestep.

It is possible to break the spatial relationship down even further. Concretely, we did examine at the spatial distribution of the highest-ranked product as a function of the lag time for rising and falling limits. We can then see that the multi-forcing LSTM learns to combine the different products in very nuanced ways, distinguishing between different memory timescales in
different basins for different hydrological conditions (Fig. 14).

## 4   Conclusions

The purpose of this paper is to show how LSTMs can leverage different precipitation products in spatiotemporally dynamic ways to improve streamflow simulations. The experiments show that there exist systematic and location- and time- specific differences between different precipitation products that can be learned and leveraged by deep learning. As might be expected,
the LSTMs tested here tended to improve hydrological simulations more when there were larger disagreement between different precipitation estimates in a given basin.

It is worth comparing these findings with classical conceptual and process-based hydrological models that treat precipitation estimate as an unique input. Current best-practice for using multiple precipitation products is to run an ensemble of hydrological models, such that each forcing data set is treated independently. Deep learning models not only have the ability to use a larger
number and variety of inputs than classical hydrology models. As a matter of fact, deep learning models do not need inputs that represent any given hydrological variable or process, and therefore have the potential to use less highly processed input data. Future work might focus on building runoff models that take as inputs the raw measurements that were used to create standard precipitation data products.

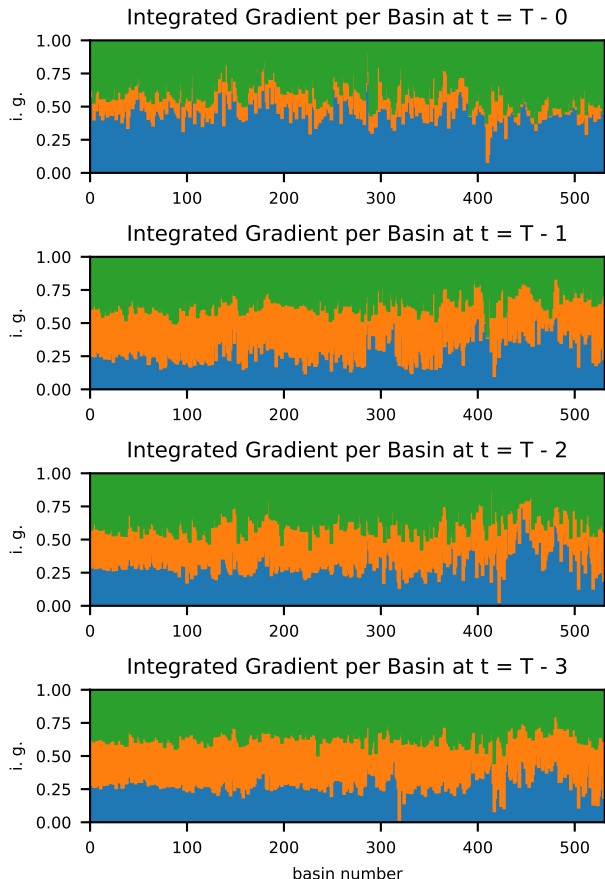

**Figure 12.** Expansion of Fig. 11 by individual basins, truncated at a lag of $s = 3$. The multi-forcing LSTM combined the precipitation products in different ways in different basins. DayMet is generally more important in high-number basins, located in the Pacific Northwest

Deep learning provides possibilities not only for improving the quality of regional (Kratzert et al., 2019b) and even un-
gauged (Kratzert et al., 2019a) simulations, but also potentially for replacing large portions of ensemble-based strategies for
uncertainty quantification (e.g. Clark et al., 2016) with multi-input models. There are many ways to deal with the uncertainty
in traditional hydrological modeling workflows. Arguably, the most common approach is to use ensembles (e.g., Clark et al.,
2016). Ensembles can be either opportunistic - i.e., from a set of pre-existing models or data products - or constructed - i.e.,
sampled from a probability distribution - (Clark et al., 2016), but in either case the idea is to use variability to represent lack
of perfect information. Multi-input deep learning has the potential to provide a fundamentally alternative method for assessing
this kind of uncertainty. Future work should additionally focus on producing predictive probabilities with multi-input deep
learning models.



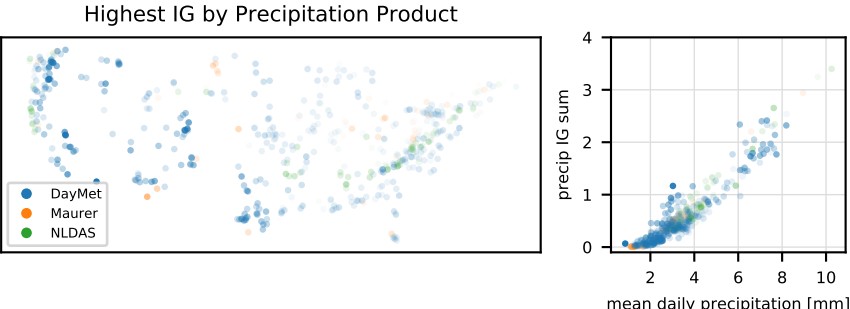

**Figure 13.** The forcing product with highest overall contribution (sensitivity) in each basin (left-hand subplot) - averaged over prediction time step and lag. The alpha value (opacity) of each dot on this map is a relative measure of the fraction of the total integrated gradients of all three precipitation products (summed over time, lag, and product) due to the highest-contributing product. The right-hand subplot shows that the total integrated gradient summed over all three precipitation products is highly correlated with total precipitation in the basin.

## 5   Code availability

The code to reproduce all results and figures will be made available at https://github.com/kratzert/multiple_forcing

## 280   6   Data availability

The validation periods of all benchmark models used in this study are available at https://doi.org/10.4211/hs.474ecc37e7db45baa425cdb4fc1b61e1. The extended Maurer forcings, including daily minimum and maximum temperature, are available at https://doi.org/10.4211/hs.17c896843cf940339c3c3496d0c1c077. The extended NLDAS forcings, including daily minimum and maximum temperature, are available at https://www.hydroshare.org/resource/285   0a68bfd7ddf642a8be9041d60f40868c/.



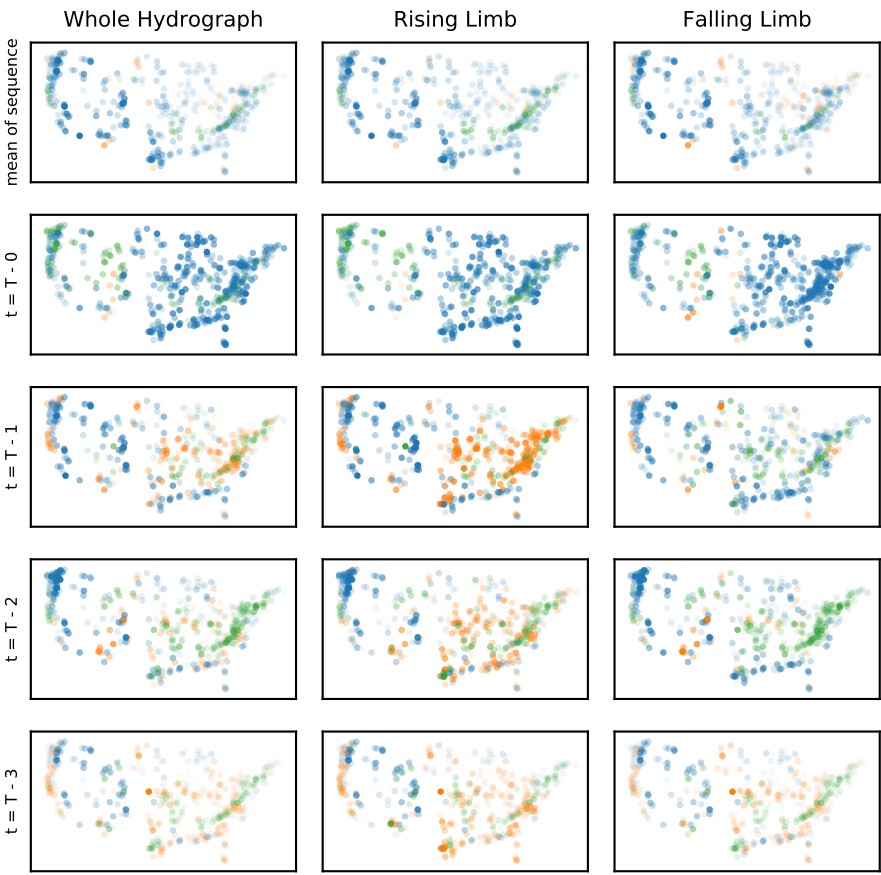

**Figure 14.** Spatial distribution of highest-ranked precipitation products at specific lags (different rows) over the whole hydrograph (left-hand column), and the rising- and falling-limbs of the hydrograph (center and right-hand columns, respectively), where blue circles denote DayMet, orange circles denote Maurer and green circles denote NLDAS. The take-away from this figure is that the multi-forcing LSTM learns to combine the different products in different ways for different memory timescales in different basins and under different hydrological conditions. The alpha value (opacity) of each dot is a relative measure of the fraction of the total integrated gradients of all three precipitation products due to the highest-contributing product.





**Table A1.** Average single LSTM performance over a variety of metrics. The average single model performances is computed as the mean of the metric of the the $n = 10$ model repetitions.

|  | NLDAS | Maurer | DayMet | Maurer + DayMet | NLDAS + Maurer | NLDAS + DayMet | All combined |
|---|---|---|---|---|---|---|---|
| NSE[i] (median) | 0.72 | 0.73 | 0.74 | 0.77 | 0.77 | 0.79 | 0.80 |
|  | ±0.003 | ±0.003 | ±0.002 | ±0.003 | ±0.004 | ±0.002 | ±0.001 |
| NSE[i] (mean) | 0.68 | 0.70 | 0.70 | 0.73 | 0.74 | 0.75 | 0.76 |
|  | ±0.003 | ±0.006 | ±0.002 | ±0.003 | ±0.002 | ±0.002 | ±0.002 |
| KGE[ii] (median) | 0.74 | 0.76 | 0.76 | 0.79 | 0.78 | 0.79 | 0.80 |
|  | ±0.006 | ±0.005 | ±0.003 | ±0.005 | ±0.008 | ±0.005 | ±0.004 |
| Pearson r[iii] (median) | 0.86 | 0.87 | 0.88 | 0.89 | 0.89 | 0.90 | 0.90 |
|  | ±0.002 | ±0.002 | ±0.002 | ±0.001 | ± 0.001 | ±0.001 | ±0.001 |
| $\alpha$-NSE[vi] (median) | 0.83 | 0.86 | 0.86 | 0.88 | 0.85 | 0.87 | 0.88 |
|  | ±0.010 | ±0.011 | ±0.008 | ±0.007 | ±0.007 | ±0.005 | ±0.008 |
| $\beta$-NSE[v] (median) | -0.03 | -0.03 | -0.03 | -0.03 | -0.03 | -0.03 | -0.02 |
|  | ±0.005 | ±0.004 | ±0.004 | ± 0.004 | ± 0.004 | ±0.002 | ±0.004 |
| FHV[vi] (median) | -17.28 | -13.89 | -15.00 | -12.52 | -14.20 | -13.15 | -11.91 |
|  | ±0.904 | ±1.217 | ±0.504 | ± 0.791 | ± 0.881 | ±0.450 | ±0.549 |
| FLV[vii] (median) | -0.88 | 2.83 | 0.05 | -4.02 | 0.86 | -1.54 | 2.57 |
|  | ± 7.637 | ±5.403 | ±6.056 | ± 6.825 | ±5.499 | ±6.955 | ±4.072 |
| FMS[viii] (median) | -9.44 | -7.31 | -5.96 | -5.60 | -7.55 | -6.93 | -6.69 |
|  | ±1.293 | ±1.500 | ± 1.234 | ±1.241 | ±1.358 | ± 0.911 | ± 1.678 |
| Peak-Timing[ix] (median) | 0.46 | 0.49 | 0.46 | 0.44 | 0.42 | 0.41 | 0.41 |
|  | ± 0.010 | ±0.009 | ±0.008 | ±0.007 | ± 0.007 | ± 0.009 | ± 0.015 |

[i]: *Nash-Sutcliffe efficiency: $(-\infty, 1]$, values closer to one are desirable.*

[ii]: *Kling-Gupta efficiency: $(-\infty, 1]$, values closer to one are desirable.*

[iii]: *Pearson correlation: $[-1, 1]$, values closer to one are desirable.*

[vi]: *$\alpha$-NSE decomposition: $(0, \infty)$, values close to one are desirable.*

[v]: *$\beta$-NSE decomposition: $(-\infty, \infty)$, values close to zero are desirable.*

[vi]: *Top 2 % peak flow bias: $(-\infty, \infty)$, values close to zero are desirable.*

[vii]: *30 % low flow bias: $(-\infty, \infty)$, values close to zero are desirable.*

[viii]: *Bias of FDC midsegment slope: $(-\infty, \infty)$, values close to zero are desirable.*

[ix]: *Lag of peak timing: $(-\infty, \infty)$, values close to zero are desirable.*





**Table B1.** Average single LSTM performance across a range of different hydrological signatures. The derived metric for each signature is the Pearson correlation between the signature derived from the observed discharge vs. the signature derived from the simulated discharge. The average single model performances is then reported as the mean value of the the $n = 10$ model repetitions.

| | NLDAS | Maurer | DayMet | Maurer + DayMet | NLDAS + Maurer | NLDAS + DayMet | All combined |
|---|---|---|---|---|---|---|---|
| Baseflow index | 0.93 | 0.92 | 0.93 | 0.94 | 0.93 | 0.93 | 0.92 |
| | ±0.014 | ±0.018 | ±0.011 | ±0.005 | ±0.013 | ±0.009 | ±0.018 |
| HFD mean | 0.95 | 0.97 | 0.97 | 0.97 | 0.97 | 0.97 | 0.97 |
| | ±0.004 | ±0.003 | ±0.002 | ±0.002 | ±0.003 | ±0.003 | ± 0.004 |
| High flow dur. | 0.82 | 0.85 | 0.83 | 0.86 | 0.85 | 0.85 | 0.85 |
| | ±0.027 | ±0.014 | ±0.010 | ±0.014 | ±0.014 | ±0.008 | ± 0.014 |
| High flow freq. | 0.82 | 0.82 | 0.82 | 0.82 | 0.81 | 0.81 | 0.79 |
| | ±0.013 | ±0.014 | ±0.016 | ±0.016 | ±0.040 | ± 0.032 | ±0.037 |
| Low flow dur. | 0.44 | 0.42 | 0.46 | 0.47 | 0.43 | 0.46 | 0.45 |
| | ±0.033 | ±0.027 | ±0.025 | ±0.035 | ±0.018 | ±0.015 | ±0.039 |
| Low flow freq. | 0.83 | 0.82 | 0.84 | 0.86 | 0.82 | 0.84 | 0.83 |
| | ±0.020 | ±0.044 | ±0.028 | ±0.022 | ±0.027 | ±0.021 | ±0.043 |
| Q5 | 0.95 | 0.95 | 0.96 | 0.96 | 0.95 | 0.96 | 0.96 |
| | ±0.005 | ±0.006 | ±0.003 | ±0.003 | ± 0.005 | ±0.005 | ±0.003 |
| Q95 | 0.99 | 0.99 | 0.98 | 0.99 | 0.99 | 0.99 | 0.99 |
| | ±0.001 | ±0.001 | ±0.001 | ±0.001 | ± 0.000 | ±0.001 | ±0.000 |
| Q mean | 0.99 | 1.00 | 0.99 | 0.99 | 1.00 | 0.99 | 1.00 |
| | ±0.001 | ±0.000 | ±0.001 | ±0.000 | ±0.000 | ±0.000 | ±0.000 |
| Runoff ratio | 0.98 | 0.98 | 0.98 | 0.98 | 0.98 | 0.98 | 0.99 |
| | ±0.002 | ± 0.001 | ± 0.001 | ± 0.001 | ± 0.001 | ±0.001 | ± 0.001 |
| Slope FDC | 0.62 | 0.63 | 0.59 | 0.56 | 0.59 | 0.59 | 0.57 |
| | ±0.095 | ±0.053 | ± 0.093 | ± 0.053 | ± 0.061 | ±0.091 | ±0.096 |
| Stream elasticity | 0.61 | 0.69 | 0.70 | 0.70 | 0.68 | 0.69 | 0.71 |
| | ±0.015 | ±0.024 | ±0.017 | ±0.018 | ±0.025 | ±0.032 | ±0.021 |
| Zero flow freq. | 0.30 | 0.42 | 0.27 | 0.33 | 0.33 | 0.31 | 0.28 |
| | ±0.101 | ± 0.097 | ± 0.088 | ± 0.080 | ±0.067 | ± 0.086 | ±0.085 |



## Appendix C:  Peak flow timing

To evaluate the model performance on the peak timing we used the following procedure: First, we determined peaks in the observed runoff time series by locality search. That is, potential peaks are defined as local maxima. To reduce the number of peaks and filter out noise, the next step was an iterative process where, by pairwise comparison, only the maximum peak is
290 kept until all peaks have at least a distance of 100 time steps to each other. The procedure is implemented in SciPy's find_peak function (Virtanen et al., 2020) and is used in the current work.

Second, we iterated over all peaks and searched for the corresponding peak in the simulated discharge time series. The simulated peak is defined as the highest discharge value inside of a window of $\pm 3$ days around the observed peak. And, the peak timing error is the offset between the observed peak and the simulated peak. The resulting metric is the average offset
over all peaks.

*Author contributions.*  FK had the idea for the training LSTMs on multiple forcing products. FK, DK, and GN designed all the experiments. FK trained the models and evaluated the results. GN did the triple collocation analysis, as well as the integrated gradients analysis. GN supervised the manuscript from the hydrological perspective and SH from the machine-learning perspective. GN and SH share the responsibility for the last authorship in the respective fields. All the authors worked on the manuscript.

*Competing interests.*  The authors declare that they have no conflict of interest.

*Acknowledgements.*  Authors from the Johannes Kepler University acknowledge support by Bosch, ZF, Google (Faculty Research Award), the NVIDIA Corporation with the GPU donations, LIT (grant no. LIT-2017-3-YOU-003) and FWF (grant no. P 28660-N31). Grey Nearing acknowledges support from the NASA Advanced Information Systems Technology program (award ID 80NSSC17K0541).

The project relies heavily on open source software. All programming was done in Python version 3.7 (van Rossum, 1995) and associated
libraries including: Numpy (Van Der Walt et al., 2011), Pandas (McKinney, 2010), PyTorch (Paszke et al., 2017), SciPy (Virtanen et al., 2020), Matplotlib (Hunter, 2007) and xarray (Hoyer and Hamman, 2017)



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
