# Peer review of "A note on leveraging synergy in multiple meteorological datasets with deep learning for rainfall-runoff modeling"

_Hydrology and Earth System Sciences, 2020_

## Referee Comment (RC1) · Anonymous Referee #1 · 22 Jun 2020

The paper describes the use of deep learning rainfall-runoff models based on Long Short Term Memory networks for combining multiple forcing products and improve the model accuracy relative to using only individual input datasets. The approach is demonstrated over 531 basins in the CAMELS dataset. Overall, the approach is technically sound, the manuscript is very well written, and the general topic is interesting for HESS readership. However, there are a few of points that I would recommend to clarify before the paper is accepted for publications.

1- the main contribution of the paper should be better contextualised with respect to the existing (and fast growing) literature on the topic. the current manuscript introduction is indeed relatively short (i.e. 30 lines) and only introduces the purpose of this study without illustrating other existing methods. while I found the idea of the proposed approach interesting, neither the use of deep learning hydrologic models or the idea of data fusion is completely new and, therefore, the paper will benefit from a critical analysis of existing methods and how the proposed model is advancing the state of the art. Moreover, I would recommend to better clarify the novel contribution of this paper wrt the sequence of previous publications by the same authors using LSTMs for rainfall-runoff models (I'm not saying this paper is not advancing the previous ones, but considering also the concerns related to the benchmarking discussed at point 2 I believe the authors should clearly demonstrate that the contribution of this paper is beyond the "minimum publication unit").

2- the set up of the benchmarking analysis is not fully convincing as the authors are comparing their model accuracy against (A) models calibrated using a single product and (B) traditional hydrologic models from Kratzert et al. (2019b). While the first analysis is the core of the paper, I don't understand the reason for the second one for two main reasons: in Kratzert et al. (2019b) the authors have already demonstrated the superiority of LSTMs wrt standard hydrologic model; if the new models that combines multiple inputs outperform the LSTMs using a single forcing as shown in (A), it comes straight that the new models also perform better than standard hydrologic models. In addition, this second benchmarking might confuse some readers who may attribute the reported improvements to the combination of inputs, whereas they are mostly due to the model structure. Rather than the comparison with traditional hydrologic models (which cannot use multiple meteo forcing data as the LSTMs), I would suggest the paper will benefit much more from a benchmarking against other state-of-the-art data driven models.

3- lastly, the paper is in my opinion a bit lengthy with 14 figures that make the narrative a bit scattered. I would then suggest to explore the option of selecting the main findings-figures worth to be discussed in the main paper (e.g. Fig. 6 and 7) and move some

content to a supplementary material.

---

## Referee Comment (RC2) · Anonymous Referee #2 · 9 Jul 2020

The main point of the paper was that one can use multiple precipitation products in a single LSTM to improve streamflow model performance. The other analyses are secondary (and also problematic), without memorable take-home messages. One of the two conclusions in the abstract is "A sensitivity analysis showed that the LSTM learned to utilize different precipitation products in different ways in different basins and for simulating different parts of the hydrograph in individual basins", which does not seem to have said much.

I think there is some value in this idea (although incremental) of fusing multiple forcing dataset using DL, but the effectiveness of DL has not been compared to other methods

and is thus out of context. Also, stacking multiple data sources as inputs is a common practice among machine learning practitioners. The procedure itself is not the novelty, although I get it that in hydrology few might have used it.

This might be a matter of personal opinion, so I ask the editor to weigh my opinion as what it is — an opinion, the paper appears too thin and too incremental for me to warrant a HESS contribution. This is based on my understanding of HESS as a premier outlet for hydrological science. When the authors published their first couple of papers, which they cited, it was novel. Now, LSTM seems to be widely used in hydrology and it is no longer novel, so the sole point become the use of multiple forcing datasets. My personal judgement is that this point alone lacks the punch needed for a HESS paper.

Beyond the main opinion, I raise some other major points below. It does occur to me many of the claims were rather casually made in this paper and need further validation. Many details were missing, and I would be worried some results are not stable.

1.The motivation seems problematic in logic. What is such a model used for? (i) Are you using it for climate change impact assessment? You are not going to have three forcing datasets which you can train the model with. None of the datasets will be available for future climate. Climate model outputs are not able to be used in supervised training like this, with daily streamflow as the target. (ii) Are you using it for flood forecasting? It does not seem like this model is optimally wired for forecasting, which in general taps into data assimilation. It is uncertain is significant value of multiple forcing would still exist in a setting with data assimilation. (iii) Are you using it for hydrologic budget analysis? With this setting, we don't even know how much rainfall has been applied in the model from a mass balance point of view. Hence, while the results may look nice, it may not have real-world use cases!

2. I wish not to see hydrology becoming a computer-science competition where an incremental change in an experiment becomes a new paper. There should be either scientific advances or methodological innovations. The results might be publishable

but as it reads now it does not look to be at HESS level.

3. The authors claimed DL is better than traditional hydrologic models at using multiple sources of information. This is not proven. One can run an ensemble of simulations with multiple forcings, as authors said around line 270. Now, I do expect DL models to outperform, because previous papers have shown that. but perhaps the difference between single-forcing and multiple-forcing will be similar to what is shown here. At least no hard evidence was provided.

4. Does the author use only multiple precipitation data or all the forcing variables? If only multiple precipitation were used, the author should clearly state this and use "multiple precipitation" instead of "multiple meteorological/forcing" in the title and main texts. If multiple datasets were used for all forcing variables, why were the analyses only executed on the precipitation in section 3.2 and 3.3? How did you know the effects were not due to other variables? If only precipitation was used, why not use other variables?

5. The benchmark scenarios with the hydrologic models are off topic. The comparison between the LSTM and these hydrologic models have been done in Kratzert et. al, 2019. Therefore, it's quite obvious that the multi-forcing model in this study would further outperform hydrologic models. Given the main topic here is to show the effective synergy of multiple meteorological forcings using deep learning model, the fair benchmark would be the ensemble forcing simulations with hydrologic models.

6. The explanation of the triple collocation method was not clearly provided in section 2.4.2. The meanings of $\alpha$, $\beta$, e were not explained in equation 1, 2 and 3. How was Equation 4 derived from the equation 2 and 3? I was confused here. Did the author first fit a log-transformed linear model (equation 2 and 3) or directly use the covariances of time series to calculate the error variance as shown by equation 4?

7. The analysis of gradients is very unclear. In section 2.4.3 and 3.3, why did the author choose the "integrated gradients" instead of the simple gradients of inputs extracted

from the network? At the time of simulation, only the actual variable value matters, not the whole range of x. And... how did you integrate it for different values in x? It wasn't clear at all. If the author chose a more complex implementation, the comparison with the original one is needed. I believe most readers will have hard time understanding what the "gradients" really means as shown in section 3.3. For example, precipitation at previous hundreds of days (i.e. the length of training instance) can all contribute to the runoff prediction at the present time step T because of the memory characteristic of the LSTM model. Did the gradient at time step T refer to the gradient of the precipitation at the present time step or the sum/average gradients of all the previous hundreds of days? More clarifications should be given to help readers understand the gradient results here. 8. Following the above comments, the gradients of inputs w.r.t. the outputs can be quite unstable for deep learning models. Were the results shown in section 3.3 based on the gradients of ensemble runs or single model? If single model was used here, the author should show the results of multiple ensemble members as well as standard deviations and demonstrate the robustness of their gradients.

9. The analysis in section 3.2 is not convincing: 1) In line 210-212, the author clearly stated that the model performed worse in basins with lower precipitation error, especially for NLDAS forcing. This is counter-intuitive, leading me to question the validity of the triple collocation method employed and what it really says. How can the author verify the validity of this method given this abnormal result? Although they have listed the locations of those abnormal basins, they did not give convincing and detailed explanations for this problem. 2) What are the differences of "$\sigma$" and "total variance" in Figure 7 and 8? More explanations are needed here to avoid confusion. 3) Line 218-219, "with higher total precipitation variance (not triple collocation error variance), indicating better performance in wetter catchments." This expression is not rigorous since the arid areas with frequent extreme events can also have large precipitation variance, such as Texas. 4) Line 220- 221, "This is not true for the other two models, where higher total variance is associated with a higher variance in model skill, indicating higher proportion of the variance due to measurement error". This is very hard to understand and needs

further supporting evidence. I can only see that the variance spread is large for basins with high NSE performance for Maurer and NLDAS from Figure 9.

5) The same problem as the point 1) in line 228 and Figure 10, the author can not draw a general conclusion and neglect those abnormal basins in Figure 7 and 10. These basins behave differently than the conclusion, so it might be your conclusion that is wrong!

Other comments:

1) Why were only 447 of the 531 basins used for the benchmark with hydrologic models (section 2.4.1)? I went back to their 2019 paper and they used 571 there, where they benchmarked against other models. Furthermore, as we discussed earlier, there is no longer a point to benchmark against default traditional models. It has been done. I would welcome a benchmark with the ensemble forcing scenario, but that was not included.

2) The paragraph starting at line 43 cited the statements in Behnke et al. (2016) and seemed quite incoherent here. It seems this paragraph should be better moved to the introduction part.

3) Figure 7 and 6 can be combined since they tell similar stories. There have been so many figures in the paper which made the paper look redundant. Really with the actual content available in this paper 4 figures would have been adequate.
* * *

---

## Author Comment (AC1) · 22 Jul 2020

The comments of the reviewer are written black, our answers in purple.

**Anonymous Referee #1**

The paper describes the use of deep learning rainfall-runoff models based on LongShort Term Memory networks for combining multiple forcing products and improve the model accuracy relative to using only individual input datasets. The approach is demonstrated over 531 basins in the CAMELS dataset. Overall, the approach is technically sound, the manuscript is very well written, and the general topic is interesting for HESS readership. However, there are a few of points that I would recommend to clarify before the paper is accepted for publications.

We want to thank Reviewer #1 for their sincere comments and suggestions. We made two major changes based on this review:

- The first was to add a new set of benchmarks related to how multiple forcings inputs are used in a traditional hydrological modeling situation.

- The second was to shorten the manuscript by moving a lot of the existing analysis to supplementary material and reorganizing the introduction to speak more clearly to the main point of the paper, which is leveraging multiple forcing products to help address challenges in existing methods.

1. The main contribution of the paper should be better contextualised with respect to the existing (and fast growing) literature on the topic. The current manuscript introduction is indeed relatively short (i.e. 30 lines) and only introduces the purpose of this study without illustrating other existing methods. while I found the idea of the proposed approach interesting, neither the use of deep learning hydrologic models or the idea of data fusion is completely new and, therefore, the paper will benefit from a critical analysis of existing methods and how the proposed model is advancing the state of the art. Moreover, I would recommend to better clarify the novel contribution of this paper wrt the sequence of previous publications by the same authors using LSTMs for rainfall-runoff models (I'm not saying this paper is not advancing the previous ones, but considering also the concerns related to the benchmarking discussed at point 2 I believe the authors should clearly demonstrate that the contribution of this paper is beyond the "minimum publication unit").

We agree with this assessment. The introduction in our original submission was short and missing a clear statement about why this manuscript is clearly advancing over previous publications. In the revised manuscript we include new introductory material that outlines challenges related to leveraging multiple inputs in traditional hydrology models as well as related literature. We expect that this, along with the added benchmarks related to these traditional methods (see answer to remark 2), will help clarify the new contribution presented in this manuscript.

2. The set up of the benchmarking analysis is not fully convincing as the authors are comparing their model accuracy against (A) models calibrated using a single product and (B) traditional hydrologic models from Kratzert et al. (2019b). While the first analysis is the core of the paper, I don't understand the reason for the second one for two main reasons:  in Kratzert et al. (2019b) the authors have already demonstrated the superiority of LSTMs wrt standard hydrologic model; if the new models that combines multiple inputs outperform the LSTMs using a single forcing as shown in (A), it comes straight that the new models also perform better than standard hydrologic models.  In addition, this second benchmarking might confuse some readers who may attribute the reported improvements to the combination of inputs, whereas they are mostly due to the model structure.  Rather than the comparison with traditional hydrologic models (which cannot use multiple meteo forcing data as the LSTMs),  I would suggest the paper will benefit much more from a benchmarking against other state-of-the-art data driven models.

We do understand why one would come to these conclusions. Nevertheless, we believe that the model comparison in Figure 4 is important, since it contextualizes and highlights the improvement we see due to using multiple inputs in a single LSTM. It gives a sense of how much this improvement really is (the multi-forcing LSTM almost - not quite - doubles the performance gap between LSTM-based models and traditional hydrological models).

We added this analysis to contextualize the results of our current manuscript to our previous studies, where we trained LSTMs just on a single forcing product. The purpose of the hydrological benchmark models is  to highlight the improvement of the model performance over single-forcing LSTMs .

However, we agree with the reviewer that including a different set of benchmarks improves the manuscript. In the revised manuscript (uploaded on invitation by the editor) we benchmarked against arguably the most common method of using multiple forcing products in the context of traditional hydrological models, which is to train separate hydrological models for each forcing product, and to combine their outputs using ensembling techniques. We used the SAC-SMA + Snow-17 model, which is used for operational forecasting in the US and was also the model originally included in the CAMELS data set. To account for stochasticity in the optimization process, we calibrated multiple models per basin and forcing (similar to what was done in the original CAMELS paper by Addor et al.). The code and simulation outputs will be made available.

Regarding adding different state-of-the-art data driven models: We are not aware of any other data-driven modeling approach (something that is not based on LSTMs) that yields similar

performance for regional/continental modeling tasks (i.e. one model that predicts discharge everywhere) and can thus also be applied to forecasting (e.g., PUB,as was shown for the LSTMs in one of our previous publications).

3. Lastly, the paper is in my opinion a bit lengthy with 14 figures that make the narrative a bit scattered. I would then suggest to explore the option of selecting the main findings-figures worth to be discussed in the main paper (e.g. Fig. 6 and 7) and move some content to a supplementary material.

Thanks for the suggestion. We agree with this assessment and thus moved a lot of material from the original manuscript to supplementary sections.

---

## Author Comment (AC2) · 22 Jul 2020

The comments of the reviewer are written black, our answers in purple.

**Anonymous Referee #2**

The main point of the paper was that one can use multiple precipitation products in a single LSTM to improve streamflow model performance. The other analyses are secondary (and also problematic), without memorable take-home messages. One of the two conclusions in the abstract is "A sensitivity analysis showed that the LSTM learned to utilize different precipitation products in different ways in different basins and for simulating different parts of the hydrograph in individual basins", which does not seem to have said much. I think there is some value in this idea (although incremental) of fusing multiple forcing dataset using DL, but the effectiveness of DL has not been compared to other methods and is thus out of context.

Thank you for acknowledging the value of our work. The ability to learn nonlinear, nonstationary and spatially adaptive mixing strategies is a 'holy grail' of ensemble modeling - we're not sure how one could characterize this particular finding as having "not said much".

We do fully accept and agree with the first reviewer's comment that this was not emphasized well enough against a backdrop of current approaches for using ensembles of inputs, and this has been rectified in the revised manuscript that we will upload conditional on invitation by the editor.

We changed the manuscript in three ways to emphasize this point more clearly: (1) Changing the introduction to highlight challenges and current strategies for using ensemble forcings, (2) adding more relevant benchmarks, and (3) moving a substantial portion of the hydrological analysis that is only tangentially related to this main point into supplementary material.

Also, stacking multiple data sources as inputs is a common practice among machine learning practitioners. The procedure itself is not the novelty, although I get it that in hydrology few might have used it. This might be a matter of personal opinion, so I ask the editor to weigh my opinion as what it is — an opinion, the paper appears too thin and too incremental for me to warrant a HESS contribution. This is based on my understanding of HESS as a premier outlet for hydrological science. When the authors published their first couple of papers, which they cited, it was novel. Now, LSTM seems to be widely used in hydrology and it is no longer novel, so the sole point become the use of multiple forcing datasets. My personal judgement is that this point alone lacks the punch needed for a HESS paper.

It seems that we disagree on the interpretation of the results on a fundamental level. We thus dedicate the next paragraphs to illustrate our point of view.

We do not make any claims about presenting a novel algorithm or any type of novel ML theory development (as a site note, we also never claimed that the LSTM is something novel in previous publications, in-fact it is almost 30 years old). To our knowledge we are the first to test the idea of using multiple forcing products as inputs in the context or large-scale, data-driven rainfall-runoff modeling, and the result of this very simple strategy was a relatively large improvement to simulation accuracy. As reviewer #1 implied, one can view this as an implicit form of input-fusing where the model learns the (nonlinear, heterogeneous) input combination/transformation itself.

These results are (1) novel in the context of hydrology, (2) currently the best large-sample daily streamflow simulation results ever published that we know of (that do not require data assimilation or auto-regression and could therefore be used in ungauged basins), and maybe more importantly (3) present a simple but effective solution for one of the classically 'hard' problems in hydrological modelling: how to combine information from multiple inputs in spatiotemporally heterogeneous ways. The fact that this is a simple (but effective) approach is a *strength*, not a *weakness*.

Beyond the main opinion, I raise some other major points below. It does occur to me many of the claims were rather casually made in this paper and need further validation. Many details were missing, and I would be worried some results are not stable.

1. The motivation seems problematic in logic. What is such a model used for? (i) Are you using it for climate change impact assessment? You are not going to have three forcing datasets which you can train the model with. None of the datasets will be available for future climate. Climate model outputs are not able to be used in supervised training like this, with daily streamflow as the target. (ii) Are you using it for flood forecasting? It does not seem like this model is optimally wired for forecasting, which in general taps into data assimilation. It is uncertain is significant value of multiple forcing would still exist in a setting with data assimilation. (iii) Are you using it for hydrologic budget analysis? With this setting, we don't even know how much rainfall has been applied in the model from a mass balance point of view. Hence, while the results may look nice, it may not have real-world use cases!

Daily streamflow simulation is one of the most common and impactful tasks in operational (surface) hydrology. This model is absolutely 'wired' for forecasting - we (Upstream Tech; company where the last author works) currently use a proprietary version of this model to produce operational forecasts at both the short term (10-day-out) and seasonal (multi-month) timescales using ensembles of weather forcing products. The setup tested here in hindcasting mode is a direct analogy of that operational model, but without proprietary products (like ECMWF weather forecasts, etc.). Our (last author's) company (Upstream Tech) currently sells the simulations made by a proprietary version of these models to public and private customers in environmental, hazard, and hydropower sectors.

These models work with several different forms of data assimilation (many of which we are running in the operational version of the models reported here), however data assimilation is not used in the majority of forecasting situations at the national or global scale (with any type of model) because data assimilation requires local streamflow observations, and the majority of forecast points in the US and in the world are ungauged. As an example, the US National Water Model has 2.X million forecast points and only ~18,000 of those are at gauge locations. Data assimilation is important, and is significantly easier with deep learning than with traditional hydrology models (although that is not the topic of this paper), but a forecasting model must work in ungauged locations as well.

2. I wish not to see hydrology becoming a computer-science competition where an incremental change in an experiment becomes a new paper. There should be either scientific advances or methodological innovations. The results might be publishable but as it reads now it does not look to be at HESS level.

There is no danger that hydrology will reduce to a computer science competition; and yet, this type of competition will definitely become a critical *part* of the future of hydrology as a discipline. Right now, the choice of model among hydrologists is guided more by lineage and affiliation than empirical evidence (Addor and Melsen, 2018). There is some argument to be made that once one masters a specific model to a high enough level a preference should be attached to it because it allows to solve problems faster and explore tasks in a deeper fashion (since it will not be necessary to learn everything from scratch). However, as of now, no universal principles exist that allow us to establish specific models for the given tasks and goals in hydrology. This means that we (hydrologists) are not doing the best job we could be doing to be an evidence-guided discipline. It is important that this kind of model competitions become *one aspect* of hydrological science. This does not mean that the discipline as a whole will be reduced to it - e.g., process hydrology will always remain an important part of what we do.

Simply put, hydrology is an applied science, and while the process-understanding component of the science is important, its purpose is ultimately to support societal applications (most of the time we are not exploring deep philosophical questions about the nature of the universe in hydrology). That said, empirical competition between models is a critical backbone of any discipline that cares about objectivity (see also the arguments made in Donoho, 2017). It is inevitable and important that part of the future of hydrological science will be machine learning, and to the extent that a journal chooses to ignore or de-emphasize this, that journal will position itself to not be a part of one of the most important emerging sub-branches of hydrology.

3. The authors claimed DL is better than traditional hydrologic models at using multiple sources of information. This is not proven. One can run an ensemble of simulations with multiple forcings, as authors said around line 270. Now, I do expect DL models to outperform, because previous papers have shown that. but perhaps the difference

between single-forcing and multiple-forcing will be similar to what is shown here. At least no hard evidence was provided.

Yes, we agree that this needs to be shown. This type of analysis and benchmark was added to the revision (see also our first answer to referee #1).

4. Does the author use only multiple precipitation data or all the forcing variables? If only multiple precipitation were used, the author should clearly state this and use "multiple precipitation" instead of "multiple meteorological/forcing" in the title and main texts. If multiple datasets were used for all forcing variables, why were the analyses only executed on the precipitation in section 3.2 and 3.3? How did you know the effects were not due to other variables? If only precipitation was used, why not use other variables?

In Line 62ff we state that we use all three forcing products as inputs, using all their 5 meteorological variables. We even say that two products (Maurer and NLDAS) do not include daily minimum and maximum temperature in the original CAMELS data set and therefore provide these variables with this publication (see L66 and data availability section). However, to make it as clear as possible, we included the following sentence in the revised manuscript:

*"We used all five meteorological variables of all three data products as inputs for our model"*.

We only looked at the influence of the three precipitation products because a) precipitation is arguably the most important variable in the process of rainfall-runoff modeling b) according to Behnke et al. (2016) there is little difference in all other meteorological variables in between these data products, c) we know from other research projects that precipitation has by far the most influence in LSTM-based rainfall-runoff models (see e.g. Frame et al., 2020), and d) nothing that we show or conclude implies that other variables are not important, but the point is to show that the LSTM learns to mix forcings in dynamically heterogeneous ways. We show this using the precipitation input.

It is trivial to perform similar analysis on all other variables (and we invite everyone to do this with the code we provide with our paper), but we believe that nothing would be gained from it and the paper would become even more bloated.

5. The benchmark scenarios with the hydrologic models are off topic. The comparison between the LSTM and these hydrologic models have been done in Kratzert et. al, 2019. Therefore, it's quite obvious that the multi-forcing model in this study would further outperform hydrologic models. Given the main topic here is to show the effective synergy of multiple meteorological forcings using deep learning model, the fair benchmark would be the ensemble forcing simulations with hydrologic models.

Referee #1 had a similar impression (see Referee #1, comment 2). We do however believe that these benchmarks convey important information. The purpose of including them was to

contextualize the increase in performance between the LSTMs from Kratzert et al. (2019) and the models of this manuscript, compared to the performance of traditional hydrology models. This is shown in Figure 4, which directly compares (1) the spread between traditional models, (2) the gap between traditional models and LSTMs, and (3) the extra improvement due to multiple forcings. It puts the value of this approach into context of the previous step-change from deep learning.

We also added several multi-input ensemble benchmarks to the revision (which we will upload upon invitation by the editor), which - the reviewer is correct - were missing from the original manuscript (see also answer to reviewer #1 and to comment #3 above).

6. The explanation of the triple collocation method was not clearly provided in section 2.4.2. The meanings of α, β, e were not explained in equation 1, 2 and 3. How was Equation 4 derived from the equation 2 and 3? I was confused here. Did the author first fit a log-transformed linear model (equation 2 and 3) or directly use the covariances of time series to calculate the error variance as shown by equation 4?

If possible we would like to keep the description as concise as it currently is. It is generally not customary to re-derive each established method used in a paper, especially if this method is around 20 years old and has been applied many times in the field of hydrology. It is customary to show the main equations so that a reader can develop an intuition about what the analysis means and how it works. Readers interested in deriving the method can refer to the first-order references in the paper, which are/were referenced throughout Section 2.4.2.

7. The analysis of gradients is very unclear. In section 2.4.3 and 3.3, why did the author choose the "integrated gradients" instead of the simple gradients of inputs extracted from the network? At the time of simulation, only the actual variable value matters, not the whole range of x. And... how did you integrate it for different values in x? It wasn't clear at all. If the author chose a more complex implementation, the comparison with the original one is needed. I believe most readers will have hard time understanding what the "gradients" really means as shown in section 3.3. For example, precipitation at previous hundreds of days (i.e. the length of training instance) can all contribute to the runoff prediction at the present time step T because of the memory characteristic of the LSTM model. Did the gradient at time step T refer to the gradient of the precipitation at the present time step or the sum/average gradients of all the previous hundreds of days? More clarifications should be given to help readers understand the gradient results here.

We believe that this is enough information to understand and interpret the result. Readers, interested in the method itself, do find the appropriate references throughout the manuscript.

Regarding why we choose integrated gradients instead of local gradients: Lines 159-164 of the original text explain it this way:

*"All neural networks (like LSTMs) are differentiable almost everywhere by design. Therefore, a gradient-based contribution analysis seems natural. However, as discussed by Sundararajan et al. (2017), the naive solution of using local gradients is not a reliable measures of sensitivity, since gradients might be flat even if the model response is heavily influenced by a particular input data source (which is not by necessity a bad property, see for example Hochreiter and Schmidhuber, 1997a). This is especially true in neural networks, where activation functions often include step-changes over portions of the input space - e.g., the sigmoid and hyperbolic tangent activation functions used by LSTMs have close-to-zero gradients at both extremes (see also: Shrikumar et al., 2016; Sundararajan et al., 2017)"*

What we tried to convey in this passage is that scholars and practitioners are aware that for this kind of analysis one should not use local gradients but e.g., something like the integrated gradient method. We believe that this is quite clearly stated in the above paragraph and linked to the related references.

Furthermore, there seems to be some misunderstandings on the applied method: The precipitation of the entire input sequence does influence the prediction. The integrated gradient signal we report for one time step t is not the gradient signal of only the inputs of time step t but the average integrated gradient signal over the entire input sequence (i.e. the previous 365 days), which is stated in L. 171:

*"We calculated the integrated gradients of each daily streamflow estimate in each CAMELS basin during the 10-year test period with respect to precipitation inputs from the past 365 days (the look-back period of the LSTM). That is, on day $t = T$, we calculated $1095 = 3 * 365$ integrated gradient values related to the three precipitation products. The relative integrated gradient values quantify how the LSTM combines precipitation products over time, over space, and also as a function of lag or lead-time into the current streamflow prediction"*

Additionally, in L. 233 in the results section, we explicitly re-iterate:

*"To reiterate from above, the integrated gradient is a measure of input attribution, or sensitivity such that inputs with higher integrated gradients have a larger influence on model outputs. Integrated gradients shown in Fig. 11 were averaged over all timesteps in the test period, and also over all basins. This figure shows the sensitivity of streamflow at time $t = T$ to each of the three precipitation inputs at times $t = T − s$ where s is the lag value on the x-axis."*

We also don't think that for the general reader, all details behind this method are important (similar to the triple collocation), as long as one understands what the output of this method is. This is explicitly stated in L. 233:

*"To reiterate from above, the integrated gradient is a measure of input attribution, or sensitivity such that inputs with higher integrated gradients have a larger influence on model outputs".*

8. Following the above comments, the gradients of inputs w.r.t. the outputs can be quite unstable for deep learning models. Were the results shown in section 3.3 based on the

gradients of ensemble runs or single model? If single model was used here, the author should show the results of multiple ensemble members as well as standard deviations and demonstrate the robustness of their gradients.

We had a lengthy discussion about this question, because it was not quite obvious to us what the reviewer meant. Generally, gradients are a perfectly robust concept (with gradients referring to the derivative of the loss/output w.r.t the network weights/inputs).

We think what the reviewer means is that deep learning models suffer a certain randomness due to different initialisations of the network parameters or stochasticity in the learning process (i.e. mini-batch sampling). Thus the question if the results are from a single model or not, because differently initialized models could learn different things. And, since we showed results from a single model (out of the 10 repetitions we trained, as reviewer #2 correctly points out), the learned behavior of the model could indeed be different (which, we believe, the statement of the reviewer hints at).

The reason for showing only a single model is that the variation between different models is negligible. That is, qualitatively all models make use of the three forcing products in a similar way. We will include a sentence to clarify this in the revised manuscript.
To validate this point, below are analogies of  Figure 11 of the manuscript derived from 5 different models.

The reason for not including error bars or reporting standard deviations is, because the absolute value of the integrated gradient method is not of real importance (which makes it harder to quantitatively compare integrated gradient results of two models). It is rather a relative measure, showing which parts in the inputs are more (or less) important for the model prediction. And as the reviewer and editor can see for the figures below, the relative behavior between all models is practically identical (i.e. Maurer forcings are practically ignored at the last time step and only gain importance afterwards, which is the message we want to tell with this figure).

[Figure]

[Figure]

[Figure]

[Figure]

[Figure]

9. The analysis in section 3.2 is not convincing:
   a. In line 210-212, the author clearly stated that the model performed worse in basins with lower precipitation error, especially for NLDAS forcing. This is counter-intuitive, leading me to question the validity of the triple collocation method employed and what it really says. How can the author verify the validity of this method given this abnormal result? Although they have listed the locations of those abnormal basins, they did not give convincing and detailed explanations for this problem.

We believe that these "non-intuitive" results are explained quite thoroughly in the manuscript already. Several paragraphs immediately following the one that the reviewer references are devoted to it.

There are several factors that act together to cause this phenomenon, but the main one is that NLDAS has some anomalies in particular basins in the Rocky Mountains that dominate this effect (illustrated in Figures 7 and 8). Additionally, *"Triple collocation measures (dis)agreement between measurement sources, rather than error variances directly"* (Line 216), and it is not always the case that one forcing product disagreeing with others is actually error. Figure 9 then shows how this agreement/disagreement is correlated with total precip in one, but not the other two, of the products. This points to a systematic difference between daymet and the other two products, that is picked up by triple collocation. The point we will eventually make based on this analysis is that the LSTM can exploit this systematic difference.

As a point in this review process, we would like to point out that we dedicated two paragraphs and three figures (Figure 7, 8, and 9) to explaining these non-intuitive results that the reviewer has mentioned. It's a little difficult to know what more we could do.

Just to reiterate, the larger point that this is ultimately supporting is that the information in multiple forcing products is complex - with both redundant and synergistic characteristics, - and the LSTM learns how to leverage at least a lot of this complex information aggregation in ways that are difficult to predict a priori from detectable features of the inputs themselves. We would have preferred to do this analysis using information theory, which would have let us use more concrete terms and more direct quantification of redundant and synergistic information, but there simply is not enough data on a per-catchment basis for stable information theory results, so we had to use log-linear analogues (i.e., Triple Collocation).

b.  What are the differences of "σ" and "total variance" in Figure 7 and 8? More explanations are needed here to avoid confusion.

This difference between total variance of a precipitation product and triple collocation error variance (sigma) is explained in Line 219 of the original manuscript, which the reviewer quotes in their next comment.

c.  Line 218-219, "with higher total precipitation variance (not triple collocation error variance), indicating better performance in wetter catchments." This expression is not rigorous since the arid areas with frequent extreme events can also have large precipitation variance, such as Texas.

An early version of the manuscript had plots that showed the correlation between precipitation variance and total precipitation, but we removed these figures because some early feedback we got from an informal reviewer was that this was common knowledge in hydrology (precip variance is highly correlated with total precip). If the editor or reviewer thinks that we should include this figure again, we can do it, however our impression was that this is an additional figure (since, as both reviewers mentioned, the number of figures is already too high).

d.  Line 220- 221, "This is not true for the other two models, where higher total variance is associated with a higher variance in model skill, indicating higher proportion of the variance due to measurement error". This is very hard to understand and needs further supporting evidence. I can only see that the variance spread is large for basins with high NSE performance for Maurer and NLDAS from Figure 9.

Since the reviewer seems to misunderstand the argument it could be that we compressed the message too much. Generally spoken, in any data product there are two basic sources of variance: variance of the true value and variance of the error. All we are saying here is that if increasing model skill is associated with increasing precip variance, then that increasing precip

variance is likely not due to measurement error. If increasing precip variance is not correlated with increasing model skill, then some of the precip variance is likely due to measurement error.

The point of this figure and analysis is to show that there is a systematic disagreement between the per-catchment error patterns in DayMet vs. the other two products, which explains the non-intuitive triple collocation results in the other two products where TC error (which is just a measure of agreement) is apparently not correlated with NSE improvements. This shows that while TC can 'see' systematic differences in the precip products, in this case it is likely that DayMet is better than the other two, and also carries more unique information than the other two.

We are trying to show uniqueness in the data sets without (unfortunately) actually being able to measure information directly (in the sense of mutual information). In the revised manuscript will add this explanation to the sentence that the reviewer quoted to make the message clearer.

      e. The same problem as the point 1) in line 228 and Figure 10, the author can not draw a general conclusion and neglect those abnormal basins in Figure 7 and these basins behave differently than the conclusion, so it might be your conclusion that is wrong!

As we mentioned above, it seems like the reviewer seems to have missed or misunderstood our explanation of these "non-intuitive" results from Figure 7 (which is the bulk of the text in this subsection and is critical to understanding these results). As stated above, we will try to make this section even clearer in the revised manuscript to avoid further confusion.

10. Other comments:
      a. Why were only 447 of the 531 basins used for the benchmark with hydrologic models (section 2.4.1)? I went back to their 2019 paper and they used 571 there, where they benchmarked against other models. Furthermore, as we discussed earlier, there is no longer a point to benchmark against default traditional models. It has been done. I would welcome a benchmark with the ensemble forcing scenario, but that was not included.

In our 2019 Benchmarking paper, which we think the reviewer is referencing, ("Towards learning universal, regional, and local hydrological behaviors via machine learning applied to large-sample datasets"), we used the same basins as in this current paper. We used 531 to train our model and 447 to benchmark (same as here). As we explained in our earlier paper (e.g. first sentence of Section 3.2), and in this paper (Line 40), the reason is that not all of the hydrological benchmark models are available for all basins. Simulations from all benchmark models are available at only 447 basins. Again, we did not run our own benchmarks - this was a community contribution effort, which was done to avoid bias in the implementation of the benchmark models.

b. The paragraph starting at line 43 cited the statements in Behnke et al. (2016) and seemed quite incoherent here. It seems this paragraph should be better moved to the introduction part.

If the reviewer/editor does not insist we would like to keep the paragraph where it is. It would be unusual to move such a paragraph to the introduction. This is a specific statement about the specific data products we are using. It is not a general statement about hydrologic data, and does not serve as motivation or background for our project - it is a very specific characterization from a previous study of the specific data products used here, and thus belongs in the section describing the data.

c. Figure 7 and 6 can be combined since they tell similar stories. There have been so many figures in the paper which made the paper look redundant. Really with the actual content available in this paper 4 figures would have been adequate

We believe that it is better to keep them separated. Figure 7 and 6 emphasize different aspects of the story (cause for lack of TC/NSE correlation vs. cause (elevation) of NLDAS anomaly). 'Looking redundant' is not a concern for us and these figures would take up the same amount of space and contain exactly the same information if referenced by the same figure number as they do separately.

**References**

Addor, N., & Melsen, L. A. (2019). Legacy, rather than adequacy, drives the selection of hydrological models. *Water Resources Research*, 55, 378– 390. https://doi.org/10.1029/2018WR022958

David Donoho (2017) 50 Years of Data Science, Journal of Computational and Graphical Statistics, 26:4, 745-766, DOI: 10.1080/10618600.2017.1384734

Frame, J., Nearing, G., Kratzert, F., & Rahman, M. (2020, July 2). Post processing the U.S. National Water Model with a Long Short-Term Memory network. https://doi.org/10.31223/osf.io/4xhac

---

## Editor Comment (EC1) · Dimitri Solomatine (Editor) · 7 Aug 2020

I find the referees' comments very useful, and the replies of authors also indicate that. They clearly present the plan of revisions. I wish them good luck in doing this.

---

## Referee Report (RR1)

**Overview**

This is a summary of the work as I have understood it. It does not need an explicit response but if there are any misunderstandings perhaps it would be worth clarifying them.

The paper seeks to answer two related questions. Firstly, whether using multiple input (forcing) datasets improves LSTM performance in rainfall-runoff modelling. Secondly, to extract insights about how the LSTM uses information in different times and places ("... in spatiotemporally dynamic ways.").

In order to answer these questions the authors completed two experiments:
1) tested the accuracy of the LSTM with different combinations of meteorological input datasets
2) tested the sensitivity of the LSTM to different precipitation inputs (as a demonstration of one meteorological variable)

The major finding was that LSTM model performances were improved with multiple meteorological inputs. Indeed, results are a further improvement on the previous benchmarks set by the authors in their 2019 study (Kratzert et al 2019). Not only were results improved, but the authors were able to show:
a) the DayMet dataset had the most information for rainfall-runoff modelling
b) the LSTM does learn to use different products in different locations, and different times. A simple example is that the LSTM learns a time-lag in the Maurer precipitation product, reproducing findings from elsewhere.

The novel contributions are as follows:
1) Showing that LSTMs can effectively be used with an ensemble of inputs, suggesting a simple method for combining different datasets without prior information of those datasets reliability.
2) Interpreting how an LSTM is able to use input information.
3) Demonstrating new state of the art results for rainfall-runoff modelling in a large sample study in the USA

The techniques used by this paper present an opportunity for the hydrological community to better understand LSTM based models. This fits neatly within the context of recent calls for studies to interpret machine learning methods (Nearing et al 2020, Beven 2020). While the techniques themselves (triple collocation analysis, integrated gradients, ablation studies) are not novel, the ability to use deep learning models to better understand hydrological datasets (or processes) is certainly a growing and important direction for this subfield of hydrology. This work demonstrates one use of these methods, and clearly meets the objectives set out in the introduction.

Overall, the research manuscript meets the aims of HESS and advances hydrological modelling in three ways:
1) by demonstrating the ability to utilize information from multiple input data sources, without a priori information about the reliability of this data.
2) by demonstrating techniques for interpreting LSTM based models.
3) by further improving rainfall-runoff model accuracy and providing competitive benchmarks for future rainfall-runoff modelling studies.

References:
Beven, Keith. "Deep learning, hydrological processes and the uniqueness of place." Hydrological Processes 34.16 (2020): 3608-3613.
Kratzert, Frederik, et al. "Towards learning universal, regional, and local hydrological behaviors via machine learning applied to large-sample datasets." Hydrology and Earth System Sciences 23.12 (2019): 5089-5110.
Nearing, Grey S., et al. "What role does hydrological science play in the age of machine learning?." Water Resources Research (2020): e2020WR028091.

**Specific comments**

I was grateful for the following:

1) The paper is focused and the structure is clear. The subsection titles signpost exactly what the reader is expecting to find.
2) The thoroughness of the description and analysis in Appendix E - Analysis of Precipitation uncertainty is great. The experiment using triple collocation analysis is very well described and overall, this section is an exemplar of a valuable appendix.
3) The comparison with both the SAC-SMA and LSTMs trained with single meteorological products was useful to demonstrate that the traditional hydrological models are not able to utilise this information as effectively as the deep learning methods. While this is not surprising it would have been easy to exclude it.
4) I was very keen to download and play with the code available on github. Indeed, I was able to download and run the models as described and am always impressed when results are made available and reproducible like this. Thank you very much!

I have some comments:

My main comment was that it was only clear to me on second reading that the model received multiple meteorological inputs rather than just multiple precipitation products. Having looked through some of the previous reviews it seems that I wasn't the only one. I understand that it has been clearly stated here: P4 L78: "*We used all five meteorological variables of all three data products as inputs for our model*". The confusion I think, stems from two places. 1) the abstract 2) the analysis with only precipitation products.

1) in the abstract you have written P1 L3-4 "*Using multiple precipitation products (NLDAS, Maurer, DayMet) in a single LSTM significantly improved simulation accuracy relative to using only individual precipitation products.*" Unless I have misunderstood, would it not be better to write that "Using **meteorological inputs from different data products** (NLDAS, Maurer, DayMet) in a single LSTM significantly improved simulation accuracy relative to using only individual meteorological products."

2) The analysis focuses on precipitation products (rightly given that it is the most important input variable). I think that you should explicitly write this somewhere, perhaps using the response that you used in your comments to reviewer 2: "*We only looked at the influence of the three precipitation products because a) precipitation is arguably the most important variable in the process of rainfall-runoff modeling b) according to Behnke et al. (2016) there is little difference in all other meteorological variables in between these data products, c) we know from other research projects that precipitation has by far the most influence in LSTM-based rainfall-runoff models (see e.g. Frame et al., 2020), and d) nothing that we show or conclude implies that other variables are not important, but the point is to show that the LSTM learns to mix forcings in dynamically heterogeneous ways. We show this using the precipitation input. It is trivial to perform similar analysis on all other variables (and we invite everyone to do this with the code we provide with our paper),*". Perhaps something like: "For the analysis that follows we only consider the sensitivity of the LSTM to precipitation inputs. This is for two reasons. 1) Precipitation is consistently found to be the most important variable for rainfall-runoff modelling (Frame et al., 2020) 2) according to Behnke et al. (2016) there is little difference in all other meteorological variables in between these data product"

This can go in BOTH or EITHER of P5 L129 Section 2.5 & P6 L243 Section 2.5.2. Just to make clear why you are only looking at sensitivity to precipitation. This should also clear up any confusion about the inputs to the LSTM for future readers.

**Formatting**

These are a list of small formatting / spelling errors.

P5 L103: Space between "(1)all" -> "(1) all"

P8 L178: Spelling "calbrated" -> "calibrated"

P12 L228: Double word "... in the left-subplot, and the the overall sensitivity" -> "in the left-subplot, and the overall sensitivity"

Appendix (need to be considered together)
- P16 L270 Re-label Table C1 -> A2 (or B1 depending on whether it needs its own subsection)
- P17 Appendix C is missing (or is Table C1 meant to be Table B2 ?)

- Appendices are shifted by 1 (B, D, E); Replace with ABC or ABCD (depending on whether Appendix C has it's own
- Figure captions updated depending upon the chosen appendix structure (e.g. If Appendix E becomes Appendix C, update Figure E1. to Figure C1)

**Suggestions**

Feel free to incorporate these or to ignore them. I have tried to offer my best suggestion for how a suggestion could be addressed in red.

P2 41-42: Feel free to ignore, but it is perhaps useful to use the same units for the two resolutions. You write: "the former has 1 km x 1 km spatial resolution and the latter two have one-eighth degree spatial resolution". Perhaps replace with: "the former has 1 km x 1 km spatial resolution and the latter two have 12.5 km x 12.5 km"

P5 L103-105: Experimental Design. It might be useful if you label this experiment "Feature Ablation", since you describe it once here and then again in Section 2.5 L125-128. That would make it clearer that Analysis 1 - Feature Ablation is the same as the experiments that you describe at the start of Section 2.4. Or else, include the following sentence in L128. "The feature ablation study describes the seven input configurations with different input datasets, discussed above", or words to that effect.

P9 L202-206 I am confused about what the difference is between two different benchmarkings. You write: "*The three-forcing LSTM outperformed the single forcing LSTMs almost everywhere. Individual exceptions where "less is more" do, however, exist (e.g., Southern California). Concretely, the three-forcing model **was better than the best single forcing model** in 66% of the basins (351 of 531) **and had a higher NSE than the individual single-forcing LSTMs** in over 80% of the basins*". What is the difference here? That the 3-forcing LSTM does better than the best single forcing LSTM in 66% of basins (n = 531), but better than 80% of all basin-feature combinations (n=531 * 3)? Perhaps the confusion comes from the non-specificity of better. I was thinking initially that "better" meant something other than improved NSE, since you explicitly write "higher NSE" in that sentence but leave it vague before.

P11 Figure 5: Is it possible to have more information in the caption. Perhaps including a description of what +ve and -ve values mean. "Positive (blue) values represent basins where the improvement of the 3-forcing LSTM over the comparison single-forcing model is larger. Negative values (brown) values reflect basins where the comparison single-forcing model outperforms the 3-forcing LSTM."

P12 Figure 6: Could you include information about which model you are using this information from? I am assuming it is the 3-forcing LSTM, but it could possibly be the learned contribution for each single-forcing LSTM for each respective product (DayMet, Maurer, NLDAS). "The

integrated gradients were calculated for the 3-forcing model (the model with all of the precipitation products used as input)"

P13 Figure 7: Would a key be useful? Or at least a description of the colours in the figure caption (probably easier). "The integrated gradient of Daymet is shown in blue, Maurer in orange and NLDAS in green." I know it's the same throughout the paper but I think it would help people to navigate the figure.

P19 Figure E1: Is it worthwhile including two more pieces of information in the caption? 1) That each point is a basin 2) Define what rho and sigma represent. 1) is definitely implicit and easy to understand in the context of the other figures in the appendix. However, this reviewer feels that it would be useful to be explicit about this, at least for this first plot in the appendix. 2) is defined in the text (Equation E4,E5), however, it might be useful to have a caption that fully describes the axes labels. Perhaps you can also describe what the values describe E.g. "$\rho$ describes how much correlation there is between the given data product and the estimated truth"; "$\sigma$ describes the estimated disagreement between the given data product and the other data products" (or something more correct along these lines!)

P22 Figure E5: Similar to above, is it possible to describe what the log-determinant of the covariance matrix describes? E.g. "$|\Sigma|$ increases when there is a larger disagreement between the three datasets, approximating the joint entropy of the three products"

---

## Author Response (AR2)

We thank the reviewer for his comments as well as text suggestions. We adapted most of them as is and for the others, we left an explanation below.

I have some comments:

My main comment was that it was only clear to me on second reading that the model received multiple meteorological inputs rather than just multiple precipitation products. Having looked through some of the previous reviews it seems that I wasn't the only one. I understand that it has been clearly stated here: P4 L78: "We used all five meteorological variables of all three dataproducts as inputs for our model". The confusion I think, stems from two places. 1) the abstract 2) the analysis with only precipitation products.

1) in the abstract you have written P1 L3-4 "Using multiple precipitation products (NLDAS, Maurer, DayMet) in a single LSTM significantly improved simulation accuracy relative to using only individual precipitation products." Unless I have misunderstood, would it not be better to write that "Using meteorological inputs from different data products (NLDAS, Maurer,DayMet) in a single LSTM significantly improved simulation accuracy relative to using only individual meteorological products."

We adapted the abstract as proposed by the reviewer.

2) The analysis focuses on precipitation products (rightly given that it is the most important input variable). I think that you should explicitly write this somewhere, perhaps using the response that you used in your comments to reviewer 2: "We only looked at the influence of the three precipitation products because a) precipitation is arguably the most important variable in the process of rainfall-runoff modeling b) according to Behnke et al. (2016) there is little difference in all other meteorological variables in between these data products, c) we know from other research projects that precipitation has by far the most influence in LSTM-based rainfall-runoff models (see e.g. Frame et al., 2020), and d) nothing that we show or conclude implies that other variables are not important, but the point is to show that the LSTM learns to mix forcings in dynamically heterogeneous ways. We show this using the

precipitation input. It is trivial to perform similar analysis on all other variables (and we invite everyone to do this with the code we provide with our paper),". Perhaps something like: "For the analysis that follows we only consider the sensitivity of the LSTM to precipitation inputs. This is for two reasons. 1) Precipitation is consistently found to be the most important variable for rainfall-runoff modelling (Frame et al., 2020) 2) according to Behnke et al. (2016) there is little difference in all other meteorological variables in between these data product"

This can go in BOTH or EITHER of P5 L129 Section 2.5 & P6 L243 Section 2.5.2. Just to make clear why you are only looking at sensitivity to precipitation. This should also clear up any confusion about the inputs to the LSTM for future readers.

We updated the description of the experiment in Sect. 2.5.

**Formatting**

These are a list of small formatting / spelling errors.

P5 L103: Space between "(1)all" -> "(1) all"

Thanks, corrected.

P8 L178: Spelling "calbrated" -> "calibrated"

Thanks, corrected.

P12 L228: Double word "... in the left-subplot, and the the overall sensitivity" -> "in the left-subplot, and the overall sensitivity"

Thanks, corrected.

Appendix (need to be considered together)

- P16 L270 Re-label Table C1 -> A2 (or B1 depending on whether it needs its own subsection)
- P17 Appendix C is missing (or is Table C1 meant to be Table B2 ?)
- Appendices are shifted by 1 (B, D, E); Replace with ABC or ABCD (depending on whether Appendix C has it's own
- Figure captions updated depending upon the chosen appendix structure (e.g. IfAppendix E becomes Appendix C, update Figure E1. to Figure C1)

Thanks, we corrected the labeling in the appendix.

**Suggestions**

Feel free to incorporate these or to ignore them. I have tried to offer my best suggestion for howa suggestion could be addressed in red.

P2 41-42: Feel free to ignore, but it is perhaps useful to use the same units for the two resolutions. You write: "the former has 1 km x 1 km spatial resolution and the latter two have one-eighth degree spatial resolution". Perhaps replace with: "the former has 1 km x 1 km spatial resolution and the latter two have 12.5 km x 12.5 km"

We used two different units, because we preferred to report the spatial resolution as reported by the original dataset producer. However, we adapted the sentence slightly to:

"...the former has 1 km x 1 km spatial resolution and the latter two have one-eighth degree (approximately 12.5 km x 12.5 km) spatial resolution."

P5 L103-105: Experimental Design. It might be useful if you label this experiment "Feature Ablation", since you describe it once here and then again in Section 2.5 L125-128. That would make it clearer that Analysis 1 - Feature Ablation is the same as the experiments that you describe at the start of Section 2.4. Or else, include the following sentence in L128. "The feature ablation study describes the seven input configurations with different input datasets, discussed above", or words to that effect.

We think that Section 2.4 (Experimental design) and Section 2.5 (Analysis) are two different things. Section 2.4 describes the experiments that we run (i.e. train LSTMs for all different combinations of meteorological forcings) and Section 2.5 describes how we analyzed the results of these experiments. Both types of analysis (Ablation and Sensitivity) were performed on the same set of experiments (from the experimental design section).

P9 L202-206 I am confused about what the difference is between two different benchmarkings.You write: "The three-forcing LSTM outperformed the single forcing LSTMs almost everywhere. Individual exceptions where "less is more" do, however, exist (e.g., Southern California). Concretely, the three-forcing model was better than the best single forcing model in 66% of the basins (351 of 531) and had a higher NSE than the individual single-forcing LSTMs in over 80% of the basins". What is the difference here? That the 3-forcing LSTM does better thanthe best single forcing LSTM in 66% of basins (n = 531), but better than 80% of all basin-feature combinations (n=531 * 3)? Perhaps the confusion comes from the non-specificity of better. I was thinking initially that "better" meant something other than improved NSE, since you explicitly write "higher NSE" in that sentence but leave it vague before.

In both cases, better is referring to higher NSE. In the first comparison, we took the highest NSE of the three single forcing LSTMs per basin and compared those values to the result of the 3 forcing LSTM. Here, the 3 forcing LSTM is better in 66 % of the cases. In the second case, we compared the 3 forcing LSTM to each single forcing model in all basins separately. Here, the 3 forcing model is better in 441 basins than the LSTM trained with DayMet forcings (83%), better in 456 basins than the LSTM trained with Maurer forcings (86%), and better in 472 basins than the LSTM trained with NLDAS (89%). We had the detailed numbers in the caption of Fig. 5, and thus shorten this information in the text to "in over 80% of the basins". We reformulated this passage to make the comparisons clearer.

P11 Figure 5: Is it possible to have more information in the caption. Perhaps including a description of what +ve and -ve values mean. "Positive (blue)

values represent basins where the improvement of the 3-forcing LSTM over the comparison single-forcing model is larger. Negative values (brown) values reflect basins where the comparison single-forcing model outperforms the 3-forcing LSTM."

Thanks, we adapted the figure caption to include qualitative information on the color scale.

P12 Figure 6: Could you include information about which model you are using this information from? I am assuming it is the 3-forcing LSTM, but it could possibly be the learned contribution  for each single-forcing LSTM for each respective product (DayMet, Maurer, NLDAS). "The integrated gradients were calculated for the 3-forcing model (the model with all of the precipitation products used as input)"

Thanks, we adapted the figure caption.

P13 Figure 7: Would a key be useful? Or at least a description of the colours in the figure caption (probably easier). "The integrated gradient of Daymet is shown in blue, Maurer in orange and NLDAS in green." I know it's the same throughout the paper but I think it would help people to navigate the figure.

Thanks, we adapted the figure caption.

P19 Figure E1: Is it worthwhile including two more pieces of information in the caption? 1) That each point is a basin 2) Define what rho and sigma represent. 1) is definitely implicit and easy to understand in the context of the other figures in the appendix. However, this reviewer feels that it would be useful to be explicit about this, at least for this first plot in the appendix. 2) is defined in the text (Equation E4,E5), however, it might be useful to have a caption that fully describes the axes labels. Perhaps you can also describe what the values describe E.g. "\rho describes how much correlation there is between the given data product and the estimated truth"; "\sigma describes the estimated disagreement between the given data product and the other dataproducts" (or something more correct along these lines!)

Thanks, we adopted these suggestions.

P22 Figure E5: Similar to above, is it possible to describe what the log-determinant of the covariance matrix describes? E.g. "|\Sigma| increases when there is a larger disagreement between the three datasets, approximating the joint entropy of the three products"

Thanks, we adopted these suggestions.